# Cell-type-specific control of secondary cell wall formation by Musashi-type translational regulators in *Arabidopsis*

Alicia Kairouani[1†], Dominique Pontier[1†], Claire Picart[1], Fabien Mounet[2], Yves Martinez[3], Lucie Le-Bot[4], Mathieu Fanuel[4,5], Philippe Hammann[6], Lucid Belmudes[7], Remy Merret[1], Jacinthe Azevedo[1], Marie-Christine Carpentier[1], Dominique Gagliardi[8], Yohann Couté[7], Richard Sibout[4], Natacha Bies-Etheve[1]*, Thierry Lagrange[1]*

[1]Laboratoire Génome et Développement des Plantes, Université de Perpignan via Domitia, CNRS, UMR5096, Perpignan, France; [2]Laboratoire de Recherche en Sciences Végétales, Université de Toulouse III, CNRS, INP, UMR5546, Castanet-Tolosan, France; [3]FRAIB-CNRS Plateforme Imagerie, Castanet-Tolosan, France; [4]Biopolymères Interactions Assemblages, UR1268 BIA, INRAE, Nantes, France; [5]PROBE research infrastructure, BIBS Facility, INRAE, Nantes, France; [6]Plateforme Protéomique Strasbourg Esplanade de CNRS, Université de Strasbourg, Strasbourg, France; [7]Université Grenoble Alpes, INSERM, UA13 BGE, CNRS, CEA, FR2048, Grenoble, France; [8]Institut de Biologie Moléculaire des Plantes, IBMP, CNRS, Université de Strasbourg, Strasbourg, France

**\*For correspondence:**
natacha.etheve@univ-perp.fr (NB-E);
lagrange@univ-perp.fr (TL)

†These authors contributed equally to this work

**Competing interest:** The authors declare that no competing interests exist.

**Abstract** Deciphering the mechanism of secondary cell wall/SCW formation in plants is key to understanding their development and the molecular basis of biomass recalcitrance. Although transcriptional regulation is essential for SCW formation, little is known about the implication of post-transcriptional mechanisms in this process. Here we report that two *bonafide* RNA-binding proteins homologous to the animal translational regulator Musashi, MSIL2 and MSIL4, function redundantly to control SCW formation in *Arabidopsis*. MSIL2/4 interactomes are similar and enriched in proteins involved in mRNA binding and translational regulation. MSIL2/4 mutations alter SCW formation in the fibers, leading to a reduction in lignin deposition, and an increase of 4-*O*-glucuronoxylan methylation. In accordance, quantitative proteomics of stems reveal an overaccumulation of glucuronoxylan biosynthetic machinery, including GXM3, in the *msil2/4* mutant stem. We showed that MSIL4 immunoprecipitates *GXM* mRNAs, suggesting a novel aspect of SCW regulation, linking post-transcriptional control to the regulation of SCW biosynthesis genes.

## eLife assessment

Secondary cell walls support vascular plants and conduct water throughout the plant body, and are **important** resources for lignocellulosic feedstocks. Here the authors present **convincing** genetic and biochemical evidence that secondary cell wall synthesis, known already to be under complex transcriptional control, is also controlled post-transcriptionally by MUSASHI-like RNA-binding proteins. These **important** results point to a new mechanism for control of secondary cell wall synthesis, which will be interesting to cell biologists and biochemists studying and attempting to manipulate plant biomass.

## Introduction

As the main constituent of the terrestrial plant biomass, lignocellulose represents a huge reservoir of fixed carbon and a renewable resource for bioproducts and energy that is essential for human usages (*Meents et al., 2018*). Lignocellulose biomass enclosed mainly within secondary cell wall (SCW) reinforces vessels for long-distance transport and provides physical properties to fibers allowing upright growth of the plants (*Meents et al., 2018*; *Scheller and Ulvskov, 2010*). SCWs are composite material made of high-molecular-weight biopolymers, including cellulose, hemicelluloses, and lignin, as well as cell wall proteins whose composition and structure can differ markedly between dicots and monocots (*Scheller and Ulvskov, 2010*; *Yokoyama and Nishitani, 2004*). Cellulose is synthesized at the plasma membrane by the cellulose synthase complex and is composed of linear chains of β-(1–4) glucans that aggregate to form highly crystalline cellulose microfibrils (*Scheller and Ulvskov, 2010*; *Yokoyama and Nishitani, 2004*). Xylans, the main hemicellulosic polysaccharides in SCW, are synthesized in the Golgi apparatus by a complex biosynthetic machinery and comprise a group of polysaccharides that share a common backbone of β-(1,4)-linked xylose (Xyl) units, but differs by the presence of side chains, whose nature is dependent on tissues and species (*Scheller and Ulvskov, 2010*). In particular, glucuronoxylan, the dominant xylan in the SCW of dicots, is notably substituted at *O*-2 position with α-D-glucuronic acid (GlcA) or (4-O-methyl)-α-D-glucuronic acid (MeGlcA) groups, and with O-acetyl moieties at C-2 or C-3 positions (*Scheller and Ulvskov, 2010*). By interacting with the cellulose and lignin, xylans contribute to the strengthening of the SCW, and are considered to be one of the main factors contributing to the biomass recalcitrance to enzymatic hydrolysis (*Yokoyama and Nishitani, 2004*; *Kumar et al., 2016*). Finally, lignin polymers are made up of H, G, and S units which are the results of the oxidation and the polymerisation of the *p*-coumaryl, coniferyl and Synapyl alcohols, respectively. These monomers, also called monolignols are synthesized on the ER surface/cytoplasm prior to transportation to the cell wall. Lignins confer stiffness, strength and hydrophobicity to the SCW (*Scheller and Ulvskov, 2010*; *Yokoyama and Nishitani, 2004*; *Kumar et al., 2016*).

The metabolic load imposed by the coordinated production of SCW biopolymers emphasizes the importance for plants of having a precise control over temporal and spatial expression of the corresponding SCW biosynthetic genes. Based on founding studies performed in the *Arabidopsis* model, it is generally recognized that the SCW synthesis is primarily controlled at a transcriptional level by a multilayer and interconnected network of transcription factors/TFs (*Taylor-Teeples et al., 2015*; *Zhang et al., 2018*). Key players of this regulatory network are related NAC (NAM- ATAF1/2-CUC) and MYB-type TFs that induce, in a redundant and combinatorial manner, the expression of SCW biosynthetic genes by binding their promoter region (*Taylor-Teeples et al., 2015*). In addition, a family of Class III homeodomain leucine zipper (HD-ZIPIII) TFs, whose members control several aspects of *Arabidopsis* development, was also shown to interact with SCW regulatory network, contributing to xylem cell specification and SCW synthesis (*Taylor-Teeples et al., 2015*; *Zhang et al., 2018*). Although this transcriptional level of control can gain complexity in perennial plants, functional orthologs of the main regulators were identified in many different species, suggesting that SCW deposition proceeds via a conserved mechanism in all vascular plants (*Myburg et al., 2014*; *Zhong and Ye, 2014*). During the last decade, engineering the SCW regulatory network to deposit modified lignin and/or improve polysaccharide composition became a promising strategy to optimize biomass processability (*Sibout et al., 2005*; *Whitehead et al., 2018*). However, despite huge progresses in understanding the regulation of SCW synthesis, the engineering of SCW regulatory network to improve biomass yield and digestibility has proven difficult and often impairs plant growth because of altered vascular tissue development (*Bonawitz and Chapple, 2013*; *Wang et al., 2016*).

Downstream of transcriptional regulations by TFs, RNA-binding proteins (RBPs) are essential post-transcriptional modulators of gene expression across all kingdoms of life (*Singh et al., 2015*). However, although a large cohort of RBPs exists in plants (*Reichel et al., 2016*), there is little information about RBP-mediated posttranscriptional regulation of SCW biosynthetic genes. So far, only few microRNAs (miRNAs) families have been implicated in the post-transcriptional regulation of genes involved in both regulatory and enzymatic aspects of SCW biosynthesis (*Zhang et al., 2018*). In particular, miR165/166 were shown to target HD-ZIPIII transcriptional regulators, having a direct impact on the transcriptional network associated with SCW biosynthesis (*Zhang et al., 2018*). In the other hand, miRNA397/857/408 were shown to affect lignin content *via* the targeting of laccase/LAC genes, which control lignin polymerization from monolignols precursors (*Wang et al., 2014*; *Zhao et al.,*

*2015*). In addition, two related tandem CCCH zinc-finger proteins, which exhibit both DNA and RNA binding abilities, have been proposed to modulate SCW formation by regulating the expression of genes associated with cell wall metabolism (*Chai et al., 2015*). Despite these mounting evidences, our current knowledge of the role of post-transcriptional regulators in SCW synthesis is still very limited.

In this work, we report that two RRM-domain containing RNA-binding proteins homologous to the animal translational regulator Musashi, Musashi-like2/MSIL2 and Musashi-like4/MSIL4, function redundantly to control various aspects of development, including the stiffness of the inflorescence stem in *Arabidopsis*. We show that RRM-dependent RNA-binding activity is essential for MSIL2/4 functions in vivo, and that MSIL2/4 interactomes are similar, being enriched in proteins involved in 3'-UTR binding and translational regulation. MSIL2/4 mutations alter the formation of SCW in the interfascicular fiber cells, leading to a reduction in lignin deposition, and a change in the decoration pattern of glucuronoxylan that is associated with an increase of 4-*O*-methylation of GlcA substituent. In accordance, quantitative mass-spectrometry-based protein analysis reveals an overaccumulation of glucuronoxylan biosynthetic machinery, including the GlucuronoXylan Methyltransferase3/GXM3, in the *msil2/4* mutant stem. We show that MSIL4 immunoprecipitates GXM3 mRNA in vivo, likely regulating its expression at a translational level. Our results demonstrate that MSILs regulate SCW synthesis in interfascicular fiber cells and point to a novel aspect of SCW regulation linking translational repression to regulation of SCW biosynthesis genes.

## Results

### Musashi-like MSIL2/4 proteins redundantly control specific *Arabidopsis* development processes

While many studies have implicated the MSIs in the control of gene expression during cellular proliferation, cell fate determination and cancer in animals (*Fox et al., 2015*; *Kudinov et al., 2017*), little is known about the presence and activity of MSI-type proteins in plants. In this work, we describe an RNA-binding protein family in *Arabidopsis*, whose members hereafter named as *MUSASHI-Like1* to *4/MSIL1-4*, share both sequence, domain organization, and model-based structural similarities with animal MSIs (*Figure 1A and B*). Phylogenetic analysis revealed that *Arabidopsis* indeed harbors a clade of seven *MSIL*-type genes with a prominent sub-clade harboring the *MSIL1-4* genes (*Figure 1B*). Notably, a neighboring sub-clade contains two genes, *RBGD2* and *RGBD4* (*Figure 1B*), that encode for heat-inducible RBPs implicated in the response to heat stress in *Arabidopsis* (*Zhu et al., 2022*). MSIL1-4 proteins share a common domain architecture consisting of two N-terminal RNA recognition motifs (RRM1 and RRM2) that are followed by a poorly conserved, intrinsically unstructured, carboxy-terminal region (*Figure 1A* and *Figure 1—figure supplement 1A*). The survey of the gene expression ARAPORT11 databases (https://araport.org/; http://fgcz-pep2pro.uzh.ch) revealed that the *MSIL1-4* genes are widely expressed in *Arabidopsis*, with *MSIL1* showing the lowest levels of expression (*Figure 1C*). Database searches further indicated that *MSIL* orthologs are widely distributed in land plants, from bryophytes to angiosperms (*Figure 1—figure supplement 1B*). To gain a mechanistic understanding of MSIL function in *Arabidopsis*, we determined the subcellular localization of these proteins by performing fluorescence microscopy on GFP-tagged versions of MSIL2 and MSIL4 (MSIL2G and MSIL4G). Both proteins showed a diffuse cytoplasmic distribution in root cells of stable *Arabidopsis* transgenic plants that contrasts with the nucleus and cytoplasmic distributions of the free GFP protein control (*Figure 1D*).

To further investigate the function of the MSILs in *Arabidopsis*, we raised specific antibodies against non-conserved regions of MSIL1-4 and characterized T-DNA insertion *msil* mutant lines that lack full-length *MSIL* mRNA and MSIL proteins as judged by RT-PCR and western blot experiments, respectively (*Figure 1—figure supplement 2A–C*). However, none of the four characterized single *msil* mutants showed any obvious phenotype (*Figure 1—figure supplement 3A* and data not shown). To assess the potential functional redundancy between *MSIL* genes, we crossed the single *msil* mutants together and characterized the six possible double mutant combinations (*Figure 1—figure supplement 3A*). Only the *msil2*-1/*msil4*-1 (*msil2/4*) double mutant showed discernable developmental abnormal phenotypes, including enlarged and curled leaves, early leaf senescence and a pendant inflorescence stem (*Figure 1E and F* and *Figure 1—figure supplement 3A–C*). To assess whether MSIL1 and MSIL3 also contribute to *Arabidopsis* development in the *msil2/4* double mutant

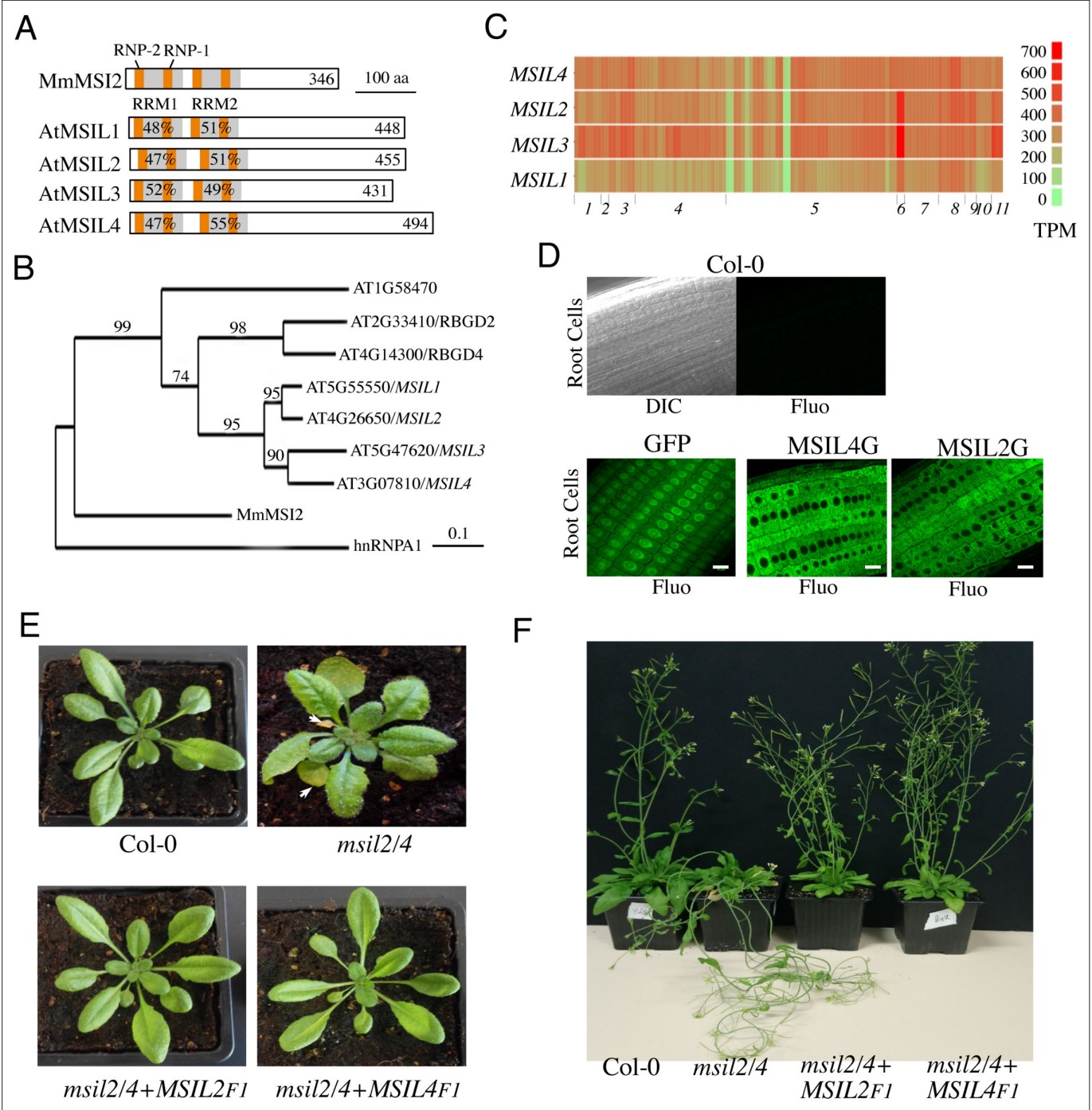

**Figure 1.** I Musashi-like MSIL2/4 proteins redundantly control development in *Arabidopsis*. (**A**) Schematic representation of the Musashi-like (MSIL) protein family in *Arabidopsis thaliana*. The invariant RNP1 and RNP2 motifs within the conserved RRM domains are indicated. Numbers refer to amino acid identities between the Mouse Musashi/MSI RRM1/2 domains and the corresponding domains in *Arabidopsis* MSIL homologs. (**B**) Evolutionary relationships between MSILs and related RNA-binding proteins. The scale bar indicates the rate of evolutionary change expressed as number of amino acid substitutions per site. (**C**) RNA-seq expression map of *MSIL* genes extracted from ARAPORT11. Shading is a $\log_2$ scale of transcripts per million (TPM). (**D**) General overview of the roots of 6-day-old wild type *Arabidopsis* seedlings (Col-0) or transgenic seedlings expressing either a free GFP protein or GFP-tagged versions of MSIL2/MSIL2G and MSIL4/MSIL4G. Scale bar, 10 μm. (**E**) Photographs of representative rosettes of Col-0, *msil2/4*, *msil2/4-MSIL2*F1 and *msil2/4-MSIL4*F1 plants. (**F**) Photographs of representative inflorescence stems of Col-0, *msil2/4*, *msil2/4-MSIL2*F1 and *msil2/4-MSIL4*F1 plants. Abbreviations: *Mus musculus* Musashi2 (MmMsi2); human heterogeneous nuclear ribonucleoprotein A1 (hnRNPA1).

The online version of this article includes the following source data and figure supplement(s) for figure 1:

**Figure supplement 1.** Organization et sequence conservation of plant MSIL proteins.

*Figure 1 continued on next page*

*Figure 1 continued*

**Figure supplement 2.** Identification of *msil* mutant lines.

**Figure supplement 2—source data 1.** Uncropped gel of semi-quantitative RT-PCR performed to characterize the msil insertion mutants.

**Figure supplement 2—source data 2.** Uncropped western gel of MSILs from either wild-type or corresponding insertion mutant.

**Figure supplement 3.** MSIL2/4 proteins redundantly control specific *Arabidopsis* development processes.

**Figure supplement 3—source data 1.** Uncropped western gel of Col-0, msil2/4, and complemented msil2/4 mutant expressing Flag-tagged versions of MSIL2 and MSIL4 proteins.

background, we generated a *msil1*-1/*msil2*-1/*msil3*-1/*msil4*-1 (*msil1/2/3/4*) quadruple mutant that shows no particular or aggravated developmental phenotypes with respect to the *msil2/4* double mutant (*Figure 1—figure supplement 3A–C*). To test the functional redundancy of MSIL2 and MSIL4 proteins in vivo, we transformed the *msil2/4* double mutant with constructs coding either for a Flag/HA-tagged version of MSIL2 (MSIL2F) or MSIL4 (MSIL4F), and selected two independent transgenic plants expressing similar and near physiological levels of the tagged protein, as evaluated by western blot using our home-made antibodies (*Figure 1—figure supplement 3D*). The ectopic expression of either MSIL2F or MSIL4F proteins was able to rescue all the phenotypes associated with the MSIL2/4 defect (*Figure 1E and F* and *Figure 1—figure supplement 3E*). This confirms that MSIL2F and MSIL4F are functional proteins and more importantly that MSIL2 and MSIL4 redundantly control various aspects of development in *Arabidopsis*.

## RRM-dependent RNA-binding activity is essential for MSIL2/4 functions in vivo

Animal MSIs interact with target transcripts through their RRM domains and mutations in these domains have been shown to impair MSI activity in vivo (*Chavali et al., 2017*; *Nguyen et al., 2020*). MSILs are very similar to metazoan MSIs over their RRM sequences (*Figure 1A* and *Figure 2—figure supplement 1A*), a closeness of sequences also observed when homology models of the MSIL4 RRM motifs were generated using SWISS-MODEL (*Biasini et al., 2014*) and AlphaFold2 (*Jumper et al., 2021*) servers (*Figure 2A* and *Figure 2—figure supplement 1B*). Homology-based structure prediction proposed that these domains adopt the characteristic RRM fold with a four-stranded β-sheet structure bearing the conserved phenylalanine residues involved in the specific recognition of RNA bases (*Daubner et al., 2013*; *Figure 2A* and *Figure 2—figure supplement 1A*). This supports the idea that MSILs have retained the ability to interact with RNA, a notion further supported by the identification of MSIL2/4 proteins in the experimentally determined *Arabidopsis* mRNA-binding proteome (*Reichel et al., 2016*). To examine the requirement for the interaction of MSIL with RNA in developmental control, we first generated an RNA-binding mutant form of MSIL4, MSIL4^RRM^, by introducing phenylalanine to aspartate (F→D) mutations in both RRM domains (*Figure 2B*). Protein-RNA interaction assays using RNA homopolymers immobilized on agarose beads confirmed that the MSIL4, but not MSIL4^RRM^, exhibited intrinsic RNA binding activity in vitro (*Figure 2C*). To assess the impact of these mutations in vivo, the MSIL4G and MSIL4G^RRM^ constructs were expressed under the control of the *MSIL4* endogenous promoter into the *msil2/4* mutant, and two independent transformants expressing near physiological levels of the WT (*MSIL4G*-3 and *MSIL4G*-8), and mutant (*MSIL4G^RRM^*-3 and *MSIL4G^RRM^*-10) proteins were selected for further complementation analysis (*Figure 2—figure supplement 2A*). MSIL4G, but not MSIL4G^RRM^ proteins were able to rescue the developmental defects incurred by *MSIL2/4* mutations (*Figure 2D* and *Figure 2—figure supplement 2B–C*), indicating that these proteins are likely to exert their functions by interacting with RNA targets in vivo.

## MSIL2/4 protein interactomes are enriched in proteins involved in 3'-UTR binding and translational regulation

Animal MSIs modulate mRNA expression mostly at a translational level through binding to conserved motifs in the 3'-UTR of the target mRNA and further interactions with various mRNA binding partners (*Fox et al., 2015*; *Weill et al., 2017*; *Cragle et al., 2019*). To further characterize the components of the MSIL2/4 network in *Arabidopsis*, we performed affinity purification coupled to LC-MS/MS, as described in *Scheer et al., 2021*, using the complemented *MSIL2F1* and *MSIL4F1* lines. Consistent with their functional redundancy, MSIL2F1 and MSIL4F1 exhibit a similar protein interaction network,

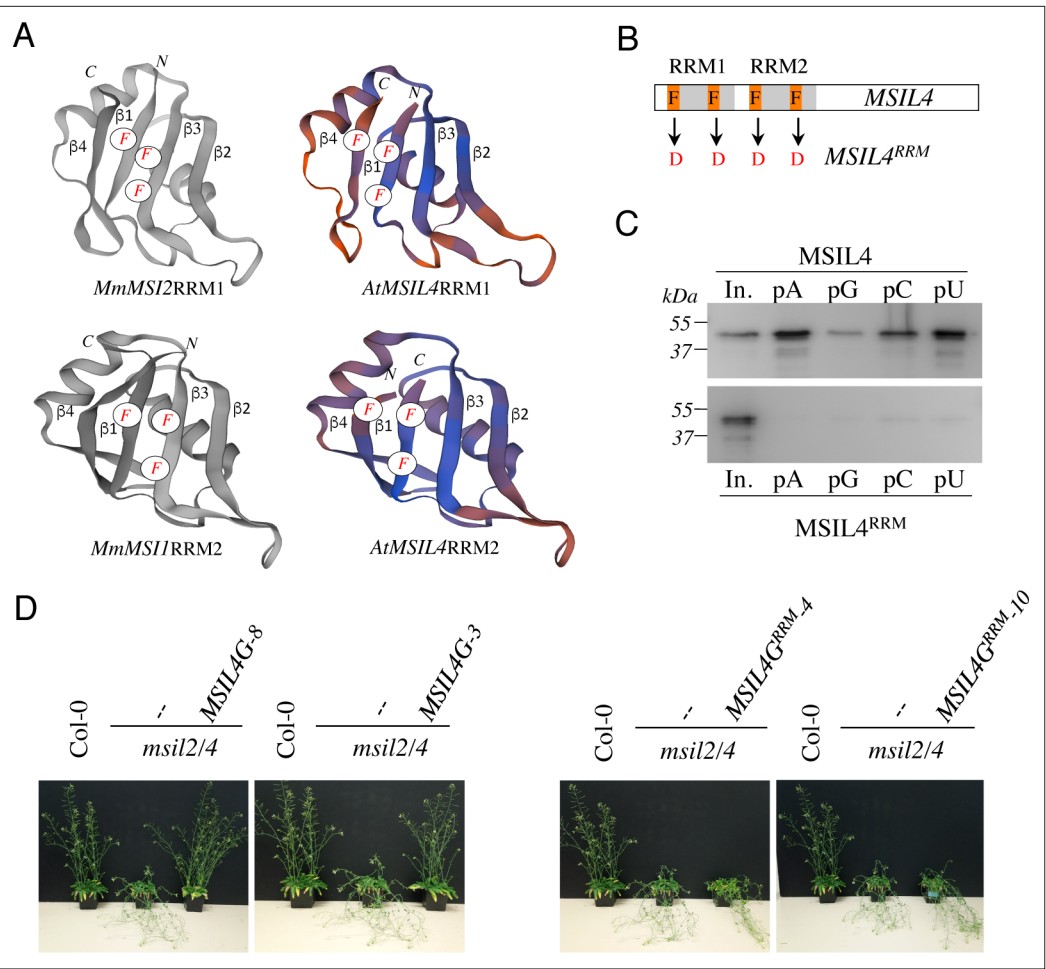

**Figure 2.** I RRM-dependent RNA-binding activity is essential for MSIL function in planta. (**A**) Experimentally determined structures of the RRM1 domain of MmMSI2 (PDB ID: 6C8U) and the RRM2 domain of MmMSI1 (PDB ID: 5x3 y), and models of the RRM domains of MSIL4 generated using the homology-modeling server SWISS-MODEL. The α-helices and β-sheets of RRM domain as well as the phenylalanine residues that contact RNA bases are highlighted. (**B**) Schematic representation of the mutations introduced in the RRM domains of MSIL4. (**C**) Binding assays of MSIL4 RRM domains on ssRNA homopolymers in vitro. Single-stranded polyA (pA), polyG (pG), polyC (pC), or polyU (pU) RNAs were subjected to binding with the His-tagged recombinant RRM domains from MSIL4 (upper part) or its mutated version (MSIL4^RRM, lower part). An anti-His antibody was used for detection. In. represents the input fraction. (**D**) RRM-dependent RNA-binding activity is essential for MSIL2/4 function in stem. Photographs of representative inflorescence stems of Col-0 and *msil2/4* mutants complemented or not with a WT version of MSIL4 (MSIL4G-8 and 3) or an rrm mutant (MSIL4G^RRM-3 and 10).

The online version of this article includes the following source data and figure supplement(s) for figure 2:

**Source data 1.** Uncropped Western gels of the binding assays of His-tagged recombinant Wild-type and mutant RRM domains.

**Figure supplement 1.** *Arabidopsis* lines expressing either wild-type or RRM-mutated forms of MSIL4 in the *msil2/4* mutant background.

**Figure supplement 2.** RRM-dependent RNA-binding activity is essential for MSIL function in vivo.

with 27 proteins significantly enriched in both MS analyses (*Figure 3A and B*, and *Supplementary file 1*). Examination of the PANTHER database (http://www.pantherdb.org) identified that MSIL2F1/4F1 are part of a protein-protein interaction network that was principally enriched in 4 categories of GO molecular functions, namely poly(A) binding protein (GO:0008143), mRNA 3-UTR binding (GO:0003730), single-stranded RNA binding (GO:0003727), and translation initiation factor activity (GO:0003743), highlighting the relationship between MSIL2/4 and the mRNA binding and regulatory proteome (*Figure 3C*).

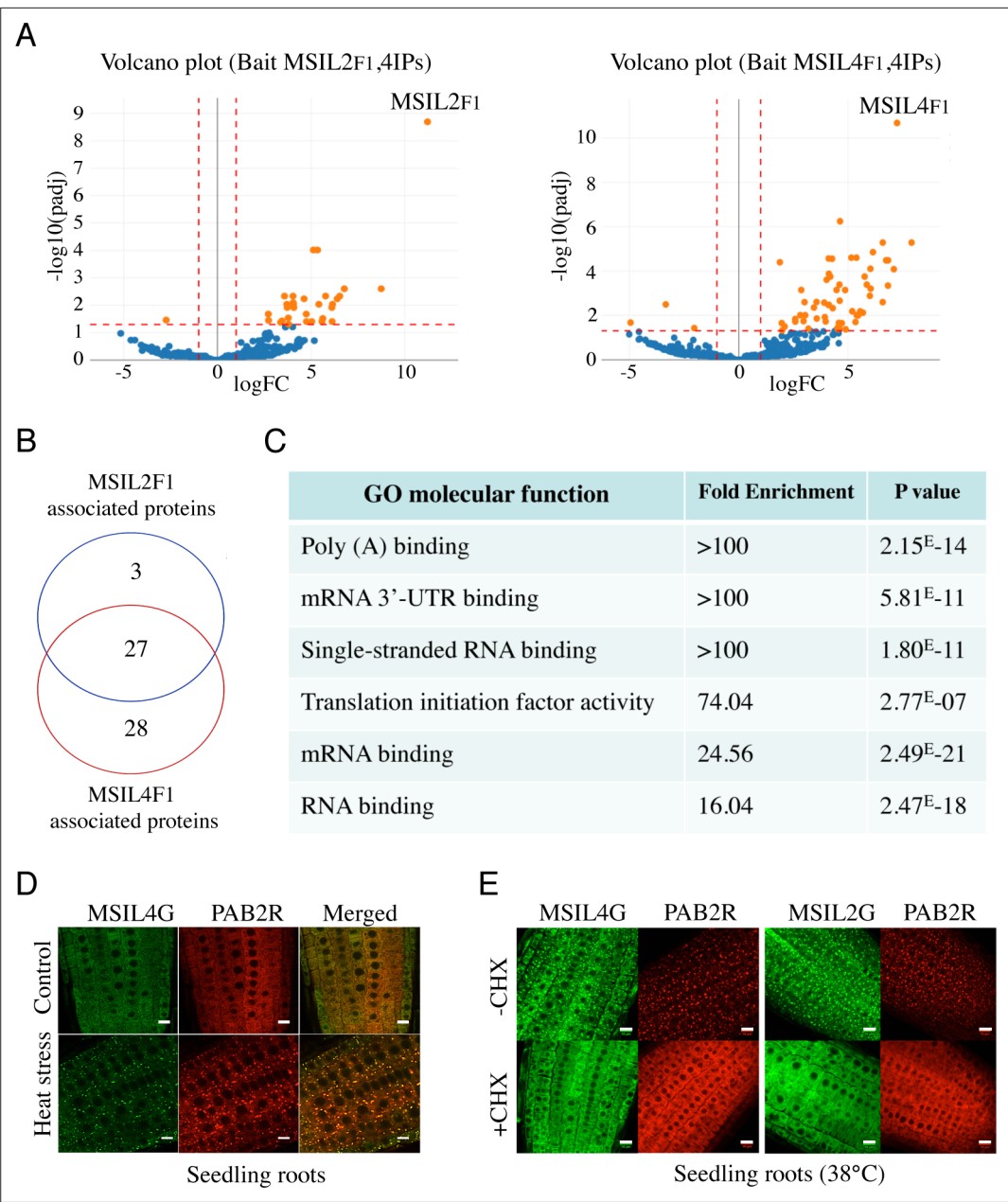

**Figure 3.** I MSIL2/4 protein interactomes are enriched in proteins involved in 3'-UTR binding and translation regulation. (**A**) The semi-volcano plots show the enrichment of proteins co-purified with MSIL2F1 and MSIL4F1 as compared with control IPs. Y- and X-axis display adjusted p-values and fold changes, respectively. The dashed lines indicate the threshold above which proteins are significantly enriched/depleted (fold change >2; adjP <0.05). (**B**) Venn diagram showing the overlap between the MSIL2 and the MSIL4 interactomes. (**C**) PANTHER-based classification of the GO molecular function that are overrepresented among the MSIL-interacting proteins. (**D**) Confocal monitoring of the colocalization of MSIL2G, MSIL4G and PAB2R in root tips of 8-d-old seedlings were monitored after 30 min of exposure to 38 °C (heat stress) or 20 °C for control treatment. (**E**) Cycloheximide treatment inhibits the formation of MSILG and PAB2R cytosolic foci. The transgenic plants expressing the stress granule marker PAB2R was used as a control. Root of 7-day-old seedlings expressing either MSIL4G, MSIL2G, or PAB2R were monitored after 1 hr of exposure to 38 °C in the presence of DMSO control treatment (DMSO) or in the presence of cycloheximide inhibitor (+CHX). Scale bar, 10 µm.

The online version of this article includes the following figure supplement(s) for figure 3:

**Figure supplement 1.** MSILs relocate to cytoplasmic granules under heat stress.

Many of the proteins present in the MSIL2/4 interactome, such as PABs, PUMs, UBP1 are involved in translation regulation and are targeted to stress granule (SGs) components, a membraneless cytoplasmic organelle that formed consecutive to the global inhibition of translation caused by heat-stress (*Chantarachot and Bailey-Serres, 2018*; *Scutenaire et al., 2018*). To assess the potential functional link between MSIL and those proteins, we crossed the MSIL2G/4 G plants with a functional PAB2-RFP/ PAB2R line (*Scutenaire et al., 2018*), and assessed the localization of the fusion proteins before and after heat-stress in both parent and crossed lines. Fluorescence analysis revealed that MSIL2G/4 G, like PAB2R, show a dispersed cytoplasmic signal under normal condition, and are recruited to cytoplasmic granules that largely overlap with PAB2R-containing SG upon heat stress as judged by the yellow-orange appearance in the merged figure (*Figure 3D* and *Figure 3—figure supplement 1*), and by an independent analysis using the Colocalization Threshold plugin from ImageJ software (data not shown). Treatment with cycloheximide (CHX), a translation inhibitor that prevent SG assembly by trapping mRNAs at polysomes (*Mahboubi and Stochaj, 2017*), confirmed that those foci correspond to bonafide SGs (*Figure 3E*). These results were consistent with the RNA-binding protein-based signature and the nature of MSIL interactomes, further supporting a functional relationship between MSILs and the translation machinery in *Arabidopsis*.

## MSIL2/4 proteins regulate the molecular architecture of SCW in *Arabidopsis* fibers

Pendant stem inflorescence is a common phenotypic trait of *Arabidopsis* mutants defective in SCW biosynthesis, that results from a lack of mechanical support and rigidity (*Zhang et al., 2018*; *Mitsuda et al., 2007*). Given the current lack of knowledge of the role of post-transcriptional mechanisms in the control of SCW biosynthesis in plants, we henceforth focused our analysis on the pendant stem phenotype of the *msil2/4* double mutant. Survey of the *Arabidopsis* inflorescence stem tissue-specific transcriptome database (*Shi et al., 2021*) indicated that the *MSIL2/4* genes are highly expressed in the SCW-forming interfascicular fiber (*if*) and xylem vessel (*xy*) cells (*Figure 4A*). Toluidine blue O staining and microscopic analysis of Col-0 and *msil2/4* mutant stem cross-sections revealed no major changes in the architecture of both the interfascicular and xylary fibers in the mutant stem. Notably, xylem cells deformation or collapse, which are a characteristic of mutants deficient in SCW synthesis, were not observed in the *msil2/4* mutant (*Figure 4B*). However, the examination of the cell wall thickness in cross sections of stems revealed that the interfascicular fiber cells display a significantly thinner SCW in *msil2/4* mutant (*Figure 4C*). UV autofluorescence and phloroglucinol staining of the Col-0, *msil2/4*, *msil2/4+MSIL2* F1, and *msil2/4+MSIL4* F1 stem sections revealed a reduced lignin signal in the interfascicular fibers of the *msil2/4* mutant compared to Col-0 or complemented plants (*Figure 4D* and *Figure 4—figure supplement 1A*). In agreement with these observations, a significant decrease of 30% in AcBr lignin content was observed in the *msil2/4* mutant compared to the Col-0 control plants (*Figure 4E*). Despite a tendency to decrease, the difference of lignin content was not significant in the complemented lines *msil2/4+MSIL2* F1 and *msil2/4+MSIL4* F1 compared to Col-0 (*Figure 4E*). To further investigate potential changes in the lignin structure and composition in the Col-0 and *msil2/4* mutant, we evaluated the impact of MSIL2/4 deficiency on lignin composition by thioacidolysis, a degradation method allowing the determination of the relative amounts of H, G and S lignin units linked by beta-O-4 linkages. Total monomer yield did not significantly differ between the Col-0 and the *msil2/4* mutant (*Figure 4—figure supplement 1B*), suggesting that the structure of the lignin polymer is not significantly affected in the mutant. Further examination of β-glucan composition using the fluorescent dye calcofluor white (*Herth and Schnepf, 1980*) revealed an increased staining in the interfascicular fiber cells of *msil2/4* mutant, supporting changes in the composition/accessibility of cellulose (*Figure 4F* and *Figure 4—figure supplement 1C*).

To extend this analysis, we then subjected purified Col-0 and *msil2/4* SCW materials to Fourier-Transform Infrared (FTIR) spectroscopy (*Alonso-Simón et al., 2011*). PLS-DA analyses performed on FT-IR absorbance data clearly separated *msil2/4* mutant lines from Col-0 plants, using the first two components (explaining 74% and 10% of total variability, respectively; *Figure 4—figure supplement 1D*, left panel). The large-scale chemotyping of CW composition in stem by FTIR spectroscopy suggested that not only lignin deposition was impaired in *msil2/4* mutants but also polysaccharides composition (*Figure 4—figure supplement 1D*, right panel; *Supplementary file 2*). The lowest absorbance values in *msil2/4* lines were observed in the region 1800–1500 cm$^{-1}$, mostly associated

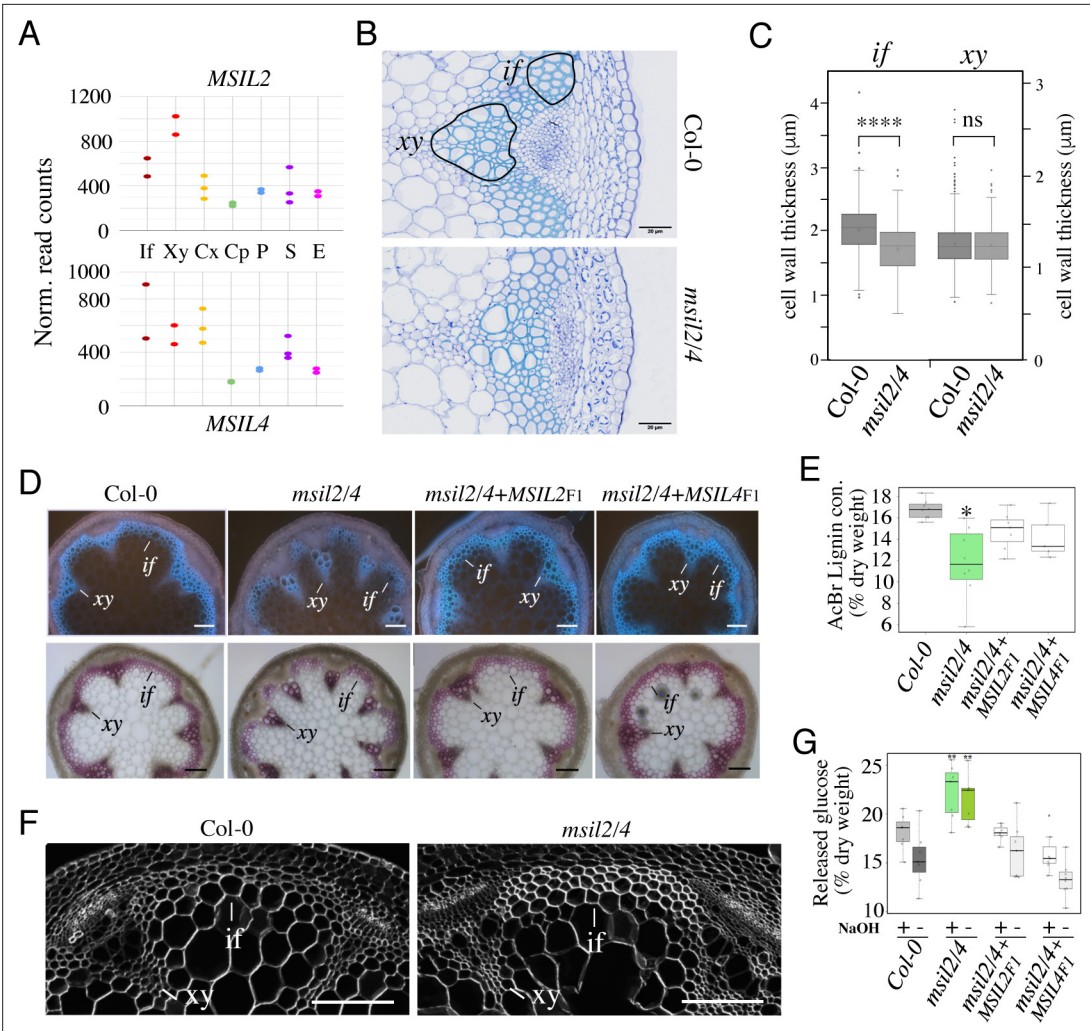

**Figure 4.** MSIL proteins regulate the molecular architecture of SCW in *Arabidopsis* fibers. (**A**) Expression profiles of the *MSIL2* and *MSIL4* genes in the inflorescence stem tissues as retrieved from the *Arabidopsis* inflorescence stem tissue-specific transcriptome database (https://arabidopsis-stem. cos.uni-heidelberg.de/). The results of two replicates are shown. F, fibers; X, xylem vessels; Cx Cambium (xylem side); Cp, Cambium (Phloem side); P, Phloem; S, Starch sheath; E, Epidermis. (**B**) Cross-sections of Col-0 and *msil2/4* stems showing vascular bundle (*xy*) and interfascicular fiber (*if*) cells stained with Toluidine blue. Representative fiber and xylary areas analyzed in the panel 4C are outlined. (**C**) Measurements of the SCW thickness of xylary fiber and interfascicular fiber cells. Values are means (n>400) ± SEM. Data were analyzed by unpaired Student's test. Asterisks indicate significant differences relative to Col-0; ****p<0.0001. (**D**) Top: lignin autofluorescence under ultraviolet (UV) using confocal microscopy in Col-0, *msil2/4* mutant, and complemented *msil2/4-MSIL2*F1 and *msil2/4-MSIL4*F1 plant stem sections. xy: xylem fibers; if: interfascicular fibers. Bottom: Phloroglucinol-HCl staining of Col-0, *msil2/4*, and complemented *msil2/4-MSIL2*F1 and *msil2/4-MSIL4*F1 plant stem sections. Scale bar, 100 μm. (**E**) Boxplots represent acetyl bromide lignin content, expressed as % of dry weight. Lignin content was measured in the bottom section of mature stems (3 biological replicates per line). Significant differences between Col-0, *msil2/4+MSIL2* F1, *msil2/4+MSIL4* F1, and *msil2/4* plant lines are indicated by asterisks*, p-value <0.01 according to Student's t test (n=6–10). (**F**) Confocal microscopy of Calcofluor-white staining of cross sections of Col-0 and *msil2/4* mutant inflorescence stems. Scale bar, 100 μm. (**G**) Boxplots represent the glucose yield after incubation of Col-0 and *msil2/4* lignocellulosic material with cellulases either without pre-treatment (-), or after pre-treatment (+) with sodium hydroxide (NaOH).

The online version of this article includes the following figure supplement(s) for figure 4:

**Figure supplement 1.** MSIL proteins regulate the molecular architecture and composition of SCW in *Arabidopsis* inflorescence stem.

to the absorbance of lignin related compounds. On the contrary, a higher absorbance was detected in *msil2/4* samples within 1050–900 cm$^{-1}$ region, mostly associated to polysaccharides compounds (**Alonso-Simón et al., 2011**). Because the optimal interaction between the SCW lignin and polysaccharide polymers is believed to contribute to SCW recalcitrance, we used saccharification as a proxy to compare cell wall sugars accessibility between Col-0, *msil2/4* and complemented mutant lines (*Figure 4G*). A higher amount of reducing sugars from native cell wall (without pre-treatment/-Tr) was

released using *msil2/4* samples compared to Col-0 samples. After cell wall loosening using alkali pre-treatment/+, an even more important amounts of sugars (+5%) was released from double mutants CW enzymatic digestion (*Figure 4G*). Together, our data indicate that MSIL2/4 regulate the molecular architecture of SCW in the interfascicular fibers, contributing significantly to the setting of biomass recalcitrance in *Arabidopsis*.

## The accumulation and activity of the glucuronoxylan decoration machinery are mis-regulated in *msil2/4* mutant

To elucidate the impact of the MSIL2/4 on the expression of SCW-related biosynthesis genes in the *Arabidopsis* inflorescence stem, we performed mRNA sequencing (mRNA-seq) on both wild-type and *msil2/4* mutant backgrounds. In total, we found 234 genes to be significantly differentially regulated (DEG) in the *msil2/4* mutant background (with stringent thresholds having a logarithm of fold change >1.5 or<−1.5 and false discovery rate FDR ≤0.05), including 156 up-regulated and 78 down-regulated mRNAs (*Figure 5—figure supplement 1A*, and *Supplementary file 3*). Interestingly, genes encoding regulatory or enzymatic components of the SCW biosynthesis pathway were absent from the DEG list, and their expression was not significant changed in *msil2/4* mutant (*Figure 5—figure supplement 1A*). Gene Ontology analysis of DEGs revealed significantly enriched signaling pathways (FDR ≤0.05) involved in plant defense or responses to biotic and abiotic stresses (*Figure 5—figure supplement 1B*), encoding products that are preferentially targeted to the extracellular region and the cell wall (*Figure 5—figure supplement 1C*).

The low representation of SCW biosynthesis genes in the DEG list suggests that MSIL2/4 could regulate the SCW formation at a post-transcriptional/translational level, a hallmark of MSI function in animal models (*Fox et al., 2015*; *Kudinov et al., 2017*). To investigate proteome-level changes that could potentially occur upon MSIL2/4 mutations, we performed a comparative proteomic analysis of Col-0 and *msil2/4* inflorescence stems using MS-based quantitative proteomics, and represented ≈4300 reliably identified and quantified proteins in a volcano plot according to their statistical p-value and their relative difference of abundance. This analysis revealed 267 proteins with significant abundances in the *msil2/4* mutant versus Col-0 inflorescence stems (fold change ≥2, Benjamini-Hochberg FDR <1%), including 156 up-regulated, and 111 down-regulated proteins (*Figure 5A* and *Supplementary file 4*). While no specific GO terms were enriched in the set of down-regulated proteins, the inspection of the up-regulated proteins revealed a strong enrichment in enzymes involved in the glucuronoxylan biosynthetic pathway (*Figure 5A and B*, and *Supplementary file 4*). Notably, the upregulated glucuronoxylan-related proteins included glycosyltransferases required for the xylan backbone synthesis (IRX9/10/14), as well as enzymes involved in its substitution (ESK1, GUX1, and GXM3) (*Scheller and Ulvskov, 2010*; *Figure 5A–C*). Visual inspection of the RNA-seq aligned reads of the up-regulated glucuronoxylan-related genes using the Integrative Genomics Viewer (IGV) and quantitative RT-PCR analysis of the glucuronoxylan-related decoration genes confirmed that the observed proteomic changes in *msil2/4* were not associated with significant variations in mRNA levels (*Figure 5—figure supplement 2A–B*).

The outcomes of our histological and proteomic analyses suggest potential variations in polysaccharide content and/or composition in the SCW of *msil2/4* mutant. To address this point, we carried out monosaccharide composition analysis on either unpretreated or acid-pretreated WT and *msil2/4* stem cell walls. The levels of glucose (Glc) and xylose (Xyl), which are the building blocks of cellulose and xylan polysaccharides in the SCW were not substantially affected in the *msil2/4* mutant (*Figure 5D*). Similarly, no major changes were observed in the levels of the primary cell wall-related monosaccharides, galactose, rhamnose, arabinose, mannose, fucose (*Figure 5D*), indicating that the MSIL2/4 mutations did not significantly altered cellulose or glucuronoxylan levels in the SCW.

To assess possible changes in glucuronoxylan decoration in *msil2/4* mutant, we performed cytochemical analysis on Col-0 and *msil2/4* stem sections using two antixylan antibodies (LM10 and LM11), whose binding specificities depend on the degree of glucuronoxylan substitution (*McCartney et al., 2006*). LM11, which binds with similar affinity both unsubstituted and substituted glucuronoxylan, shows a similar signal in the interfascicular and xylary fibers of Col-0 and *msil2/4* mutant stems, confirming that the glucuronoxylan content was not significantly modified in *msil2/4* compared to Col-0 (*Figure 5E*). In contrast, LM10, which binds less efficiently the substituted glucuronoxylan (*McCartney et al., 2006*), shows a significant signal decrease in the interfascicular fibers of *msil2/4*

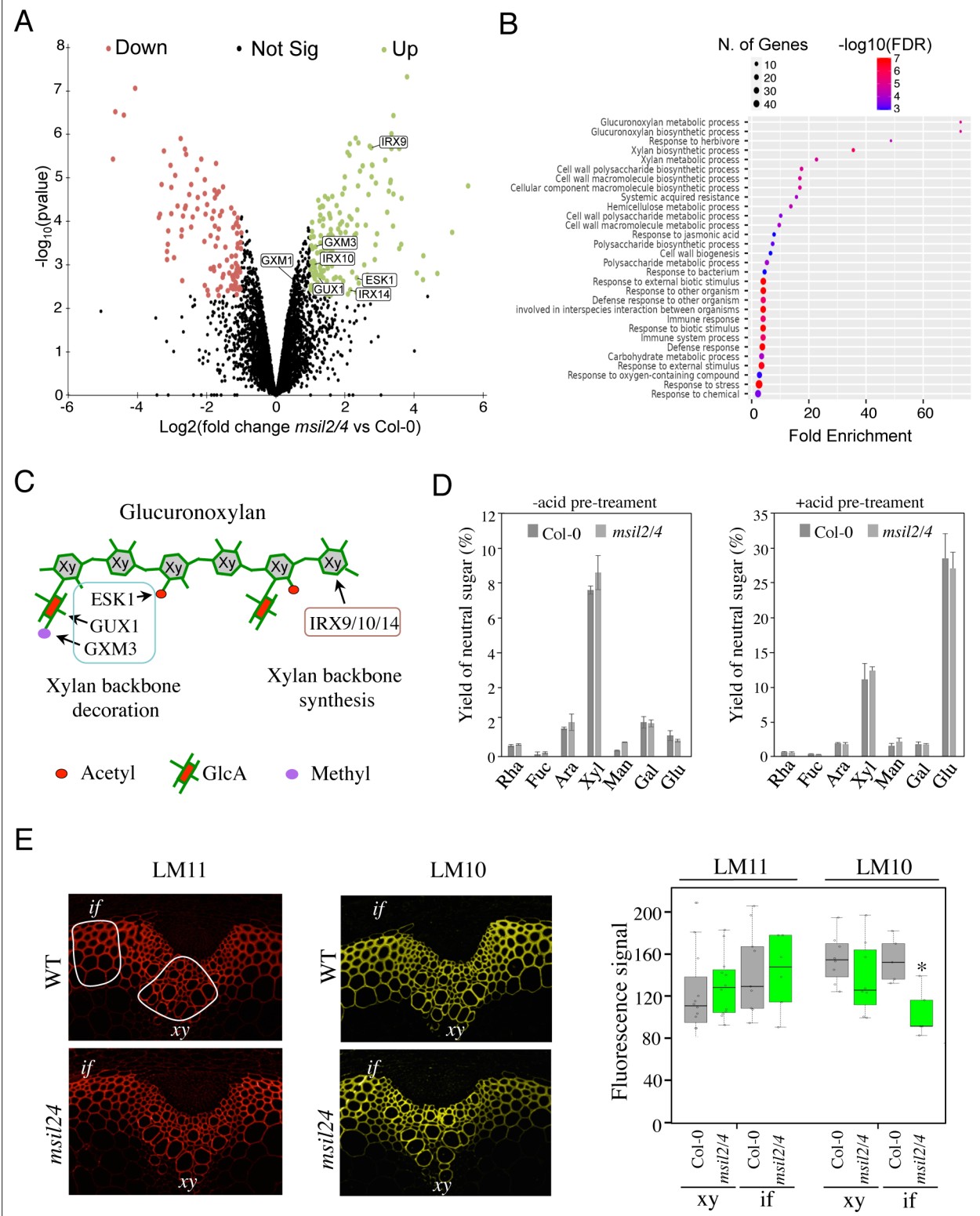

**Figure 5.** IThe accumulation and activity of the glucuronoxylan decoration machinery are altered in *msil2/4* mutant. (**A**) MS-based quantitative comparison of Col-0 and *msil2/4* inflorescence stem proteomes. Volcano plot displaying the differential abundance of proteins in Col-0 and *msil2/4* cells analyzed by MS-based label-free quantitative proteomics. The volcano plot represents the -log10 (limma p-value) on y axis plotted against the log2 (Fold Change msil2/4 vs Col-0) on x axis for each quantified protein. Green and red dots represent proteins found significantly enriched respectively in *msil2/4* and Col-0 *Arabidopsis* stems (log2(Fold Change)≥1 and -log10(p-value)≥2.29, leading to a Benjamini-Hochberg FDR = 1.01 %). The up-regulated

*Figure 5 continued on next page*

Figure 5 continued

proteins involved the glucuronoxylan biosynthetic pathways are indicated. (**B**) Gene Ontology (GO) analysis of the proteins that are significantly up-regulated in the *msil2/4* mutant. The gene ontology analysis of DEGs was performed using ShinyGO v0.76 software. Lollipop diagrams provide information about GO fold enrichment, significance (FDR in log10), and number of genes in each pathway. (**C**) Schematic model of glucuronoxylan substitution patterns in *Arabidopsis* and the enzymes involved. IRX9/10/14, glycosyltransferases involved in the synthesis of xylan (Xy) backbone. ESK1, eskimo1. GUX1, glucuronic acid substitution of xylan1; GXM3, glucuronoxylan methyltransferase3 are involved in the glucuronoxylan decoration. (**D**) Neutral monosaccharide composition of alcohol insoluble residue (AIR) extracted from inflorescence stems of wild-type and *msil2/4* mutant plants that have been pre-hydrolyzed or not with acid. Rha, rhamnose; Fuc, fucose; Ara, arabinose; Xyl, xylose; Man, mannose; Gal, galactose; Glc, glucose. (**E**) Left: Immunofluorescence labeling of transverse sections of Col-0 and *msil2/4* stems with the LM10 and LM11 antixylan antibodies. Representative fiber and xylary regions analyzed by immunofluorescence are outlined. Xy, xylem fibers; if, interfascicular fibers. Right: Quantification of the fluorescence was done using ImageJ software and processed according to **Supplementary file 5**. Significant differences are indicated by asterisks*, p-value <0.01 according to Student's t test (n=5–12).

The online version of this article includes the following figure supplement(s) for figure 5:

**Figure supplement 1.** GO analysis of the *msil2/4* DEGs reveals molecular functions associated to plant defense and response to biotic and abiotic stresses.

**Figure supplement 2.** Gene expression of glucuronoxylan-related genes in Col-0 versus *msil2/4* mutant plants.

mutant (**Figure 5E** and **Supplementary file 5**). Together, our data indicate that the MSIL2/4 mutations impact the accumulation and activity of the glucuronoxylan decoration machinery in the interfascicular fiber cells of *Arabidopsis* stem.

## MSILs restrain the degree of 4-O-methylation of glucuronoxylan in *Arabidopsis*

In *Arabidopsis*, the glucuronoxylan backbone is decorated predominantly by acetyl, GlcA, and MeGlcA groups, that are deposited in a regular and controlled manner by a specific enzymatic machinery (**Scheller and Ulvskov, 2010**; **Grantham et al., 2017**; **Wierzbicki et al., 2019**). Interestingly, the ESK1, GUX2, and GXM3 enzymes, which are involved in the deposition of the acetyl and GlcA groups, as well as the methylation of GlcA group, respectively, are over-accumulated in the *msil2/4* stem proteomics, providing a possible explanation for the modification in glucuronoxylan decoration observed in the interfascicular fiber cells of the *msil2/4* mutant. To investigate further the variations in glucuronoxylan decoration incurred by the *msil2/4* mutation, we analyzed the patterns of xylanase-released SCW oligosaccharide using matrix-assisted laser desorption ionization-time-of-flight mass spectrometry (MALDI-TOF) (**Ropartz et al., 2011**). Consistent with previous observations (**Mortimer et al., 2010**; **Urbanowicz et al., 2012**), MALDI-TOF analysis of Col-0 glucurononoxylan oligosaccharides showed the release of xylo-oligosaccharide pairs, evenly spaced by their degrees of polymerization and acetylation, bearing a GlcA (m/z 1705.5/1747.5) or methylated GlcA (m/z 1719.5/1761.5) substitution group (**Figure 6A**). Interestingly, the level of GlcA branched xylo-oligomers was strongly decreased in the *msil2/4* double mutant, and this whatever the complexity of the oligomers considered (**Figure 6A** and **Figure 6—figure supplement 1A**). Importantly, the molecular phenotypes observed in the *msil2/4* mutant were rescued in both *msil2/4+MSIL2* F1 and *msil2/4+MSIL4* F1 complemented lines, confirming the specificity of the mutation (**Figure 6A**). In agreement with the MALDI–TOF outcome, quantification of the acidic sugar levels in unpretreated (-), or acid pretreated (+) WT and *msil2/4* stem samples confirmed that the release of GlcA moiety was strongly reduced in *msil2/4* double mutant, as opposed to the galacturonic acid/GalA group located on pectins (**Figure 6B**).

It has been previously proposed that the intermediate level of glucuronoxylan methylation observed in *Arabidopsis* is due to an intrinsic rate-limiting activity of GXM enzymes in vivo (**Yuan et al., 2014**). We noticed that in addition to GXM3, a second member of the GXM family, GXM1 (**Yuan et al., 2014**), tends also to be over-expressed in *msil2/4* stem although at a fold change of 1.77 slightly below the fixed significance threshold in the proteomic analysis (**Figure 5A** and **Supplementary file 4**). This observations make us wonder whether the apparent decrease of GlcA branched xylooligo-saccharides in *msil2/4* could be due to an overmethylation of the GlcA substituent. To address this question, we performed the xylanase digestion and MALDI-TOF analysis in presence of an added spike-in oligosaccharide control (Pentaacetyl-chitopentaose) for normalization of the MALDI_TOF data (**Figure 6—figure supplement 1A**). Upon normalization, the MALDI-TOF data indicated that the decrease of the peaks corresponding to the GlcA-branched xylooligosaccharides was correlated with

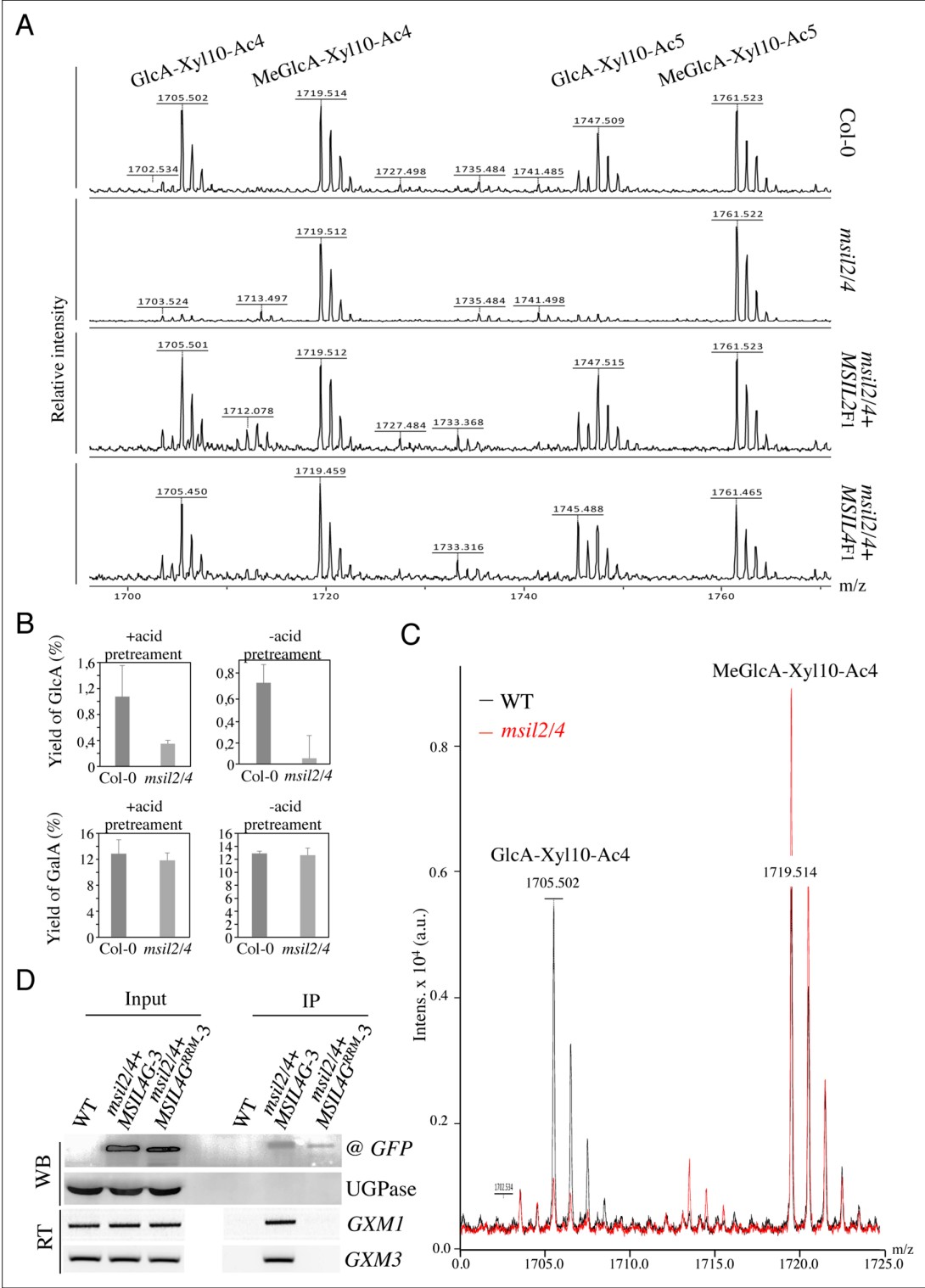

**Figure 6.** IMSILs restrain the degree of 4-O-methylation of glucuroxylan in *Arabidopsis* inflorescence stem. (**A**) MALDI-TOF mass spectra of xylooligosaccharides generated by xylanase digestion of xylan from Col-0, *msil2/4*, and complemented *msil2/4-MSIL2*F1 and *msil2/4-MSIL4*F1 inflorescence stem materials. The ions at m/z 1705/1747 and 1719/1761 correspond to acetylated xylo-decapolysaccharides bearing a GlcA residu (GlcA-Xyl10-Ac4//GlcA-Xyl10-Ac5) or a methylated GlcA residue (MeGlcA-Xyl10-Ac4//MeGlca-Xyl10-Ac5). (**B**) Acidic monosaccharide composition of alcohol insoluble residue (AIR) extracted from inflorescence stems of wild-type and *msil2/4* mutant plants that have been pre-hydrolyzed or not with acid. GalA, galacturonic acid; GlcA, glucuronic acid. (**C**) Relative changes in unmethylated/methylated GlcA decapolysaccharide ratio in Col-0 (black) and *msil2/4* mutant (red) as controlled by the addition of external spike-in control corresponding to a pentaacetyl-chitopentaose. (**D**) GFP-based RNA-IP assays. Top

*Figure 6 continued on next page*

*Figure 6 continued*

panel (WB): western blots performed using antiGFP or antiUGPase antibodies on protein fractions from inputs or antiGFP immunoprecipitates from WT and the *msil2/4* mutant plants expressing the WT (MSIL4G-3) or RRM mutant (MSIL4G^RRM-3) MSIL4-GFP fusions. Bottom panel (RT): Corresponding RT-PCR using *GXM1* or *GXM3* specific primers on RNA fractions.

The online version of this article includes the following source data and figure supplement(s) for figure 6:

**Source data 1. Uncropped Western gels of MSIL4-GFP,** MSIL4-GFP^RRM **and UGP proteins from inputs or antiGFP immunoprecipitates**.

**Source data 2.** Uncropped gel of semi-quantitative RT-PCR performed on inputs or antiGFP immunoprecipitates.

**Figure supplement 1.** MALDI-TOF mass spectra analysis of glucuronoxylan digestion products of Col-0, *msil2/4*, *msil2/4*+MSIL2 F1 and *msil2/4*+MSIL4 F1 plants.

a increase in corresponding meGlcA branched xylooligosaccharides, (*Figure 6C* and *Figure 6—figure supplement 1B*, *Supplementary file 6*), suggesting that the MSIL2/4 mutations unleash the activity of the GXM1/3 in stem. RNA immunoprecipitation assays further confirmed that the *GXM3* and *GXM1* mRNAs are efficiently pulled-down from MSIL4G-3, but not MSIL4G^RRM-3 plant extracts (*Figure 6D*), suggesting that the MSIL2/4 interaction network controls the 4-*O*-methylation of glucuronoxylan by repressing the activity of the GXM1/3 mRNAs in stem.

## Discussion

RNA-binding proteins (RBPs) are essential components of the gene regulation machinery that govern the fate and expression of cellular RNA at post-transcriptional levels, including processing, splicing, base modification, translation and degradation (*Singh et al., 2015*). Sequence-based bioinformatic, reverse and forward genetic, and affinity-based proteomic analyses have converged to show that plants, like animals and fungi, harbor a large diversity of RBPs whose functions in gene regulation and plant development are ill-defined (*Reichel et al., 2016*). In this study, we have identified an RRM motif-containing RBP family, hereafter named as MSIL, whose members share sequence, structural and functional similarities with the animal translational regulator Musashi/MSI. We found that two of the four MSIL members, MSIL2 and MSIL4, function redundantly to control various aspects of *Arabidopsis* development, including leaf senescence, morphology of rosette leaves, and the rigidity of the inflorescence stem. Our data suggest that the pleiotropy associated with the *msil2/4* mutation likely results from the multifunctionality of these proteins in *Arabidopsis* development, a situation similar to the complex phenotypic traits previously associated to MSI mutations in animals (*Fox et al., 2015*; *Kudinov et al., 2017*; *Nguyen et al., 2020*). Although the molecular bases underlying the *msil2/4* phenotypic variation remain unclear, the presence of MSILs in the high confidence *Arabidopsis* mRNA-binding proteome (*Reichel et al., 2016*), suggests that these proteins are likely to exert control over development by targeting a complex repertoire of mRNAs. Regardless of the mechanism, our data support a cell/tissue-context specific role for MSIL and provide a resource for further studies on the mechanisms of MSIL regulation in plants.

Our data also provide evidence that the RRM domains of the MSIL2/4 proteins exhibit RNA-binding ability and are essential for the activity of these proteins in vivo, an observation that is corroborated by the presence of the MSIL2/4 proteins in an experimentally determined *Arabidopsis* mRNA-binding proteome (*Reichel et al., 2016*). Further supporting a role for MSIL2/4 in mRNA-related transactions, affinity purification-mass spectrometry analysis revealed that these proteins share a common network of protein-protein interactions with well-known mRNA-binding proteins, including the PAB2/4/8, RBP45/47, and the translation factor EIF4G. Notably, the animal polyA-binding protein/PABP (homolog of the plant PABs) was previously shown to be a functional partner of MSI that is essential for it activity in translation repression (*Fox et al., 2015*; *Cragle et al., 2019*). Interestingly, the MSIL2/4-associated proteins also include the phylogenetically related RBGD2/4 proteins, two constitutively expressed RBPs that are involved in heat resistance in *Arabidopsis* (*Zhu et al., 2022*). The comparison of the MSIL2/4 and RBGD2/4 interactomes indicates only a partial overlap, suggesting a certain level of subfunctionalization. In accordance with this observation, the MSILs and RBGDs clearly diversify in function, since the *rbgd2/4* double mutant exhibits no developmental phenotypes under normal growth conditions.

A penetrant developmental phenotype associated with the MSIL2/4 mutations is a loss in stem rigidity that we could trace back to a defect in SCW formation, an important reservoir of fixed carbon and a renewable resource of major environmental and economic importance (*Meents et al., 2018*). Yet, with the exception of few microRNAs involved in the control of lignin polymer synthesis (*Zhang et al., 2018*; *Wang et al., 2014*; *Zhao et al., 2015*), no information about RBPs-mediated posttranscriptional regulation of SCW biosynthetic genes was previously reported. In this study, we show that MSIL2 and MSIL4, function redundantly to promote SCW formation in the interfascicular fibers and that their ability to bind RNA is essential to fulfill this function. Our conclusions are supported by independent approaches which highlight changes of SCW thickness, lignin content, and xylan decoration profile in the interfascicular fiber cells of the *msil2/4* mutant. The fiber-specific effect of the MSIL2/4 mutations on SCW formation, is further supported by our evidence showing that the *msil2/4* mutant grows normally and lacks the xylem collapses usually observed in more pleiotropic SCW deficient mutants. Previous studies have shown that specific master transcriptional regulators, such as NST1 and NST3, control cell-lineage-specific formation of SCW formation in *Arabidopsis* (*Taylor-Teeples et al., 2015*; *Mitsuda et al., 2007*). Our study extends this observation by showing that RBP-mediated post-transcriptional mechanisms are also involved in cell-type-specific control of SCW formation in *Arabidopsis*. Notably, the presumed role of MSIL2/4 in SCW formation in the interfascicular fiber cells contrasts with the rather constitutive expression of the corresponding *MSIL2/4* genes, suggesting as previously discussed, that MSIL2/4 activities depend on cell-context. This is in accordance with the reported cell-type specific activities of the ubiquitous MSI2 in cell renewal, cell differentiation and cancer control in animals (*Fox et al., 2015*; *Kudinov et al., 2017*). Understanding how the activity of the *MSIL2/4* genes is controlled by the cellular context will be essential from a basic point of view, providing a unique model for studying RBP regulation in specific cell type in plants.

From the set of genes found to be misregulated in the *msil2/4* stem transcriptome, none could be functionally associated with SCW biosynthesis. In contrast, we found that the MSIL2/4 mutation upregulates a battery of plant defense-related genes, known to be involved in systemic acquired resistance (SAR). In this regard, the constitutive activation of defense pathways has been well documented in mutant plants defective in cell wall formation, including the SCW (*Ellis et al., 2002*; *Miedes et al., 2014*; *Liu et al., 2023*), an observation that tends to support the idea that the transcriptome changes incurred by the MSIL2/4 disruption could be indirect. This hypothesis was supported by several lines of evidence: (1) the level of activation of defense genes in *msil2/4* is low compared to that usually observed in plants developing SAR or in mutant plants showing an auto-activation of plant immunity (*Li et al., 2010*; *Wu et al., 2012*); (2) SAR-dependent activation of defense genes is controlled mostly at a transcriptional level (*Hussain et al., 2018*); (3) the stem phenotypes displayed by *msil2/4* mutant have not been reported, to our knowledge, in the numerous plant-pathogen studies published so far; (4) the top enriched proteins in the stem proteome of *msil2/4* are players of the SCW biosynthesis pathway but not plant-defense signaling. That being said, we can't exclude that the activation of immune-related genes, that include many cell wall-modifying enzymes, additively contributes to the *msil2/4* stem phenotype. Further studies will be required to clarify this issue.

Our data reveal that the accumulation of the mRNA of SCW biosynthesis enzymes and regulators are unaffected in the *msil2/4* mutant, suggesting that the MSIL2/4 proteins act at a translational level to regulate the SCW formation. In this regard, the stem proteomics gives us a glimpse into the changes in protein abundance that could account for the observed *msil2/4* SCW phenotype. Indeed, our MS data reveal that proteins related to the glucuronoxylan biosynthesis machinery, including enzymes involved both in the synthesis (IRX9/10/14) and substitution (GUX1, ESK1, and GXM3) of the xylan backbone, are significantly over-accumulated in the *msil2/4* mutant. Knowing that the synthesis of the structurally complex glucuronoxylan polysaccharide requires the coordinated production and action of these enzymes in the Golgi apparatus, our results suggests that MSIL2/4 is exerting a multi-level control on the glucuronoxylan pathway, a mechanism reminiscent of the multi-nodal control displays by MSI2 on the TGF-β and translation pathways in animal cells (*Park et al., 2014*; *Duggimpudi et al., 2018*).

From a mechanistic point of view, however, our data raise several intriguing questions. In particular, the extent of glucuronoxylan-related protein abundance changes in *msil2/4* is at odds with the minor changes observed to glucuronoxylan composition. In particular, while the proteomic changes can correlate with the observed increase in glucuronoxylan methylation, it is not associated as expected

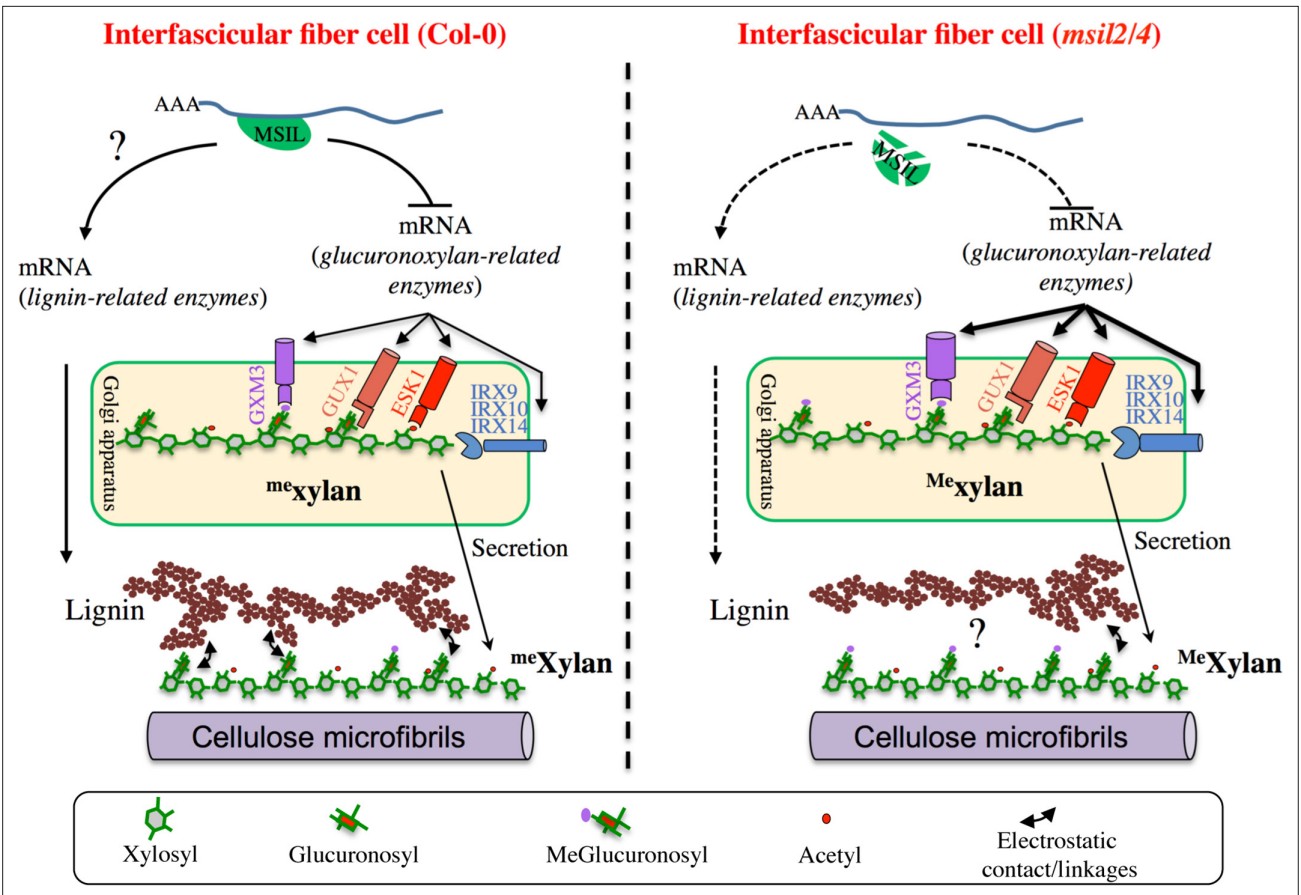

**Figure 7.** I Model of MSIL-dependent control of glucuronoxylan methylation in *Arabidopsis* and its consequence for SCW architecture. In the interfascicular fiber cells of Col-0, MSIL2/4 restrain the translation of the glucuronoxylan biosynthesis enzymes, including the rate-limiting GXM3 enzyme. This activity would keep the level of glucuronoxylan methylation at an intermediate level, therefore providing a biochemical environment that favors the interactions between the glucuronoxylan and lignin polymers. In the interfascicular fiber cells of *msil2/4* mutant, the translation of the glucuronoxylan biosynthesis machinery, including GXM3, is increased, leading to the deposition of an over-methylated form of glucuronoxylan that would be less prone to interact with lignin and establish SCW formation. In a non exclusive manner, MSIL2/4 could also have a positive role in lignin synthesis, whose defect in *msil2/4* would lead to a decrease in lignin content. The model was inspired from *Grantham et al., 2017*.

with an increase in global glucuronoxylan content. One possible explanation for this observation could be that the activities of the GUX, ESK, IRXs glucuronoxylan biosynthesis enzymes, in contrast to that of GXM3, are not rate-limiting for glucuronoxylan synthesis in vivo. Alternatively, in light with prior studies that have shown that the steady-state levels of glucuronoxylan and cellulose are correlated within the SCW (*Wu et al., 2009*; *Ye and Zhong, 2022*), one could imagine that mechanisms acting downstream of glucuronoxylan de novo synthesis step contribute to fine tune the final content in glucuronoxylan in the SCW. Comparing the levels of glucuronoxylan produced in the Golgi apparatus versus the ones deposited in the SCW in both WT ve*r*sus msil2/4 plants will be necessary to ascertain this hypothesis. In contrast, the concomitance between the glucuronoxylan hypermethylation and the GXM3 overaccumulation phenotypes observed in *msil2/4* clearly support the previous assumption that GXM-mediated methylation is a rate-limiting step in glucuronoxylan that can be overcome by GXM protein overexpression (*Yuan et al., 2014*). In fine, our data support the idea that transcriptional and post-transcriptional regulation act in a concerted manner to fine tune the GXM3 and glucuronoxylan methylation levels in the stem. We currently posit that MSIL2/4 act in addition to the transcriptional network to adjust the final level of GXM enzymes by regulating the translation of the *GXM* mRNAs (*Figure 7*). Future biochemical and functional studies will be necessary to precisely understand the mechanism by which MSIL2/4 proteins impact gene expression during SCW synthesis in *Arabidopsis*.

Our data indicate that, in addition to the reported change in glucuronoxylan methylation, the MSIL2/4 mutations is also associated with a specific decrease of lignin content in the SCW of the

interfascicular fiber cells, whose origin remains unclear. On the basis of the previous observations, we envision two possible mechanisms by which the MSIL2/4 mutations could impact the lignin deposition in the fiber cells. In the first model, we propose that MSIL2/4 positively control the lignin synthesis pathway in the interfascicular cells by promoting the translation of lignin biosynthesis genes (*Figure 7*). In this regard, although no GO enrichment could be observed for lignin biosynthesis related genes in the proteomic analysis, two lignin biosynthetic enzymes, CAD4 and PAL2, are significantly down-accumulated in *msil2/4* mutant (*Supplementary file 4*), an observation that could account for the observed decrease in lignin content. However, previous published biochemical and genetic analyses do not support this conclusion, as the single *cad4* and *pal2* knock-out mutants display no significant alteration of lignin content due to genetic redundancy (*Sibout et al., 2005*; *Rohde et al., 2004*). Moreover, thioacidolysis analyses showed that the ratio or degree of polymerization of the different lignin monomers remain unchanged in the *msil2/4* mutant, reinforcing the idea that the lignin biosynthesis pathway is functional in this mutant, leaving open the question of the specificity of MSIL2/4 action on the lignin pathway (*Figure 7*). An alternative explanation for the decrease in lignin phenotype observed in the *msil2/4* mutant would be that it results from an indirect cascade effect primarily triggered by the *msil2/4*-dependent changes in glucuronoxylan methylation. In this regard, it has been proposed that the 4-*O*-methylation of GlcA substituent results in the xylan being more hydrophobic, possibly impacting its interaction with lignin and the capacity of lignin to assemble in the SCW (*Urbanowicz et al., 2012*; *Busse-Wicher et al., 2016*; *Kang et al., 2019*; *Terrett and Dupree, 2019*; *Figure 7*). However, a 35 S promoter-dependent overexpression study of GXMs in *Arabidopsis* previously failed to report any discernable developmental or molecular phenotypes associated to a glucuronoxylan hypermethylation (*Yuan et al., 2014*). In the light of these results and despite it is necessary to elucidate deeper the role of MSIL on other SCW genes, our work provides evidence that they are new players in the regulation of SCW and may consequently impact biomass digestibility, with possible implications for improving feedstock for biorefinery.

## Methods

### Plant material

*Arabidopsis thaliana* Col-0 and mutants *msil1-1* (*GABI_462* A04), *msil2-1* (*Salk_066670*), *msil3-1* (*Salk_002477*), and *msil4-1* (*Salk_094167*) obtained from the SALK Institute genomic analysis laboratory were used in this study. PAB2-RFP/PAB2R line has been provided by Cécile Bousquet-Antonelli (LGDP, Université Perpignan). The primers used to genotype lines are listed in *Supplementary file 7*. Double mutants and quadruple mutants were generated by crossing plants. Plants were either grown in soil or cultivated in vitro on plates containing 2.20 g/l synthetic Murashige and Skoog (MS) (Duchefa) medium and 0.5 g/l MES, pH 5.7, and 7 g/l agar. To breakdown dormancy, seeds were incubated for 48 hr at 6 °C in the dark. Germination and development were performed in growth chambers, at 20 °C, 60–75% hygrometry with a 16 hr light/8 h dark photoperiod (100 µE m −2 s−1 light [fluorescent bulbs with white 6500 K spectrum, purchased from Sylvania]). For in vitro culture, the seeds were surface-sterilized before been sown on plates, incubated for 48 hr at 6 °C in the dark, and placed in a growth cabinet at 20 °C with a 16-h-d/8-h-dark cycle and 120 µE m −2 s−1 light (LEDs with white 4500 K spectrum, purchased from Vegeled).

### Transgenic lines construction

MSIL2F- and MSIL4F-tagged versions were produced from Phusion-generated genomic DNA PCR products (promoter and coding region) using primers TL976(SalI)-TL924(PstI) for MSIL2F and TL974(SalI)-TL975(BamHI) for MSIL4F. PCR products were cloned in the CTL235 binary vector containing a Flag-HA tag in front of a NosT terminator and the hygromycin resistance gene driven by a 35 S promoter. To obtain the MSIL2G and MSIL4G constructs used for fluorescence microscopy, the same PCR products were cloned in the CTL579 binary vector in front of a EGFP tag. The obtained plasmids were then used to transform *mgl2/4* plants via Agrobacterium transformation. MSIL4-GFP/MSIL4G-tagged version used for complementation and RIP corresponds to the fusion of a genomic PCR product containing *MSIL4* promoter (primers TL3527(HindIII)-TL3528(SalI)) fused with a second PCR cDNA fragment (primers TL3529(SalI)-TL3530(BamHI)) cloned initially in pGEM T Easy (Promega). After sequencing the fusion DNA has been cloned in the binary vector CTL579 containing GFP cDNA

sequence. The MSIL4G-tagged construct was finally introduced in *msil2/4* plants via Agrobacterium transformation. The rrm mutated MSIL4G^RRM version was produced as MSIL4G except that the cDNA fragment has been gene-synthetized (Genecust) to modify RNA binding ability by having aspartic instead of phenylalanine at position 9, 49 and 51 in RRM1 domain and in position 111, 151 and 153 in RRM2 domain. Transgenic seeds selection was performed in the presence of 30 µg/l hygromycin and the progenies were screened by PCR.

## Protein extraction and immunodetection

Total plant extracts (up to 100 mg) were ground in liquid nitrogen and proteins were extracted according to *Hurkman and Tanaka, 1986*. Before migration, SDS–PAGE loading buffer was added and Coomassie staining was used to calibrate loadings. Proteins were separated on SDS/ PAGE gels and blotted onto Immobilon-P PVDF membrane (Cat. No. IPVH00010; MerckMillipore). Protein blot analysis was performed using the Immobilon Western Chemiluminescent HRP Substrate (Cat. No. WBKLS0500; MerckMillipore). Specific MSIL antibodies were raised in rabbits against defined epitopes Epitope MSIL1 (SCDGTSSTFGYNRIPS), MSIL2 (RLQEYFGKYGDLVE), MSIL3 (GYGVKPEVRYSPAVGN) and MSIL4 (TWRSPTPETEGPAPFS) (Eurogentec).

## Protein production and purification

The MSIL4 RRM binding domains were amplified using primers TL4128 and TL4129 either from MSIL4G construct for the wild type version or from MSIL4GΔr for the mutated version. PCR products were then cloned in pET41 to obtain His-tagged RRM domains. Tagged proteins were produced by induction in *Escherichia coli* BL21 cells. Cultures of 200 mL of an ampicillin-resistant (100 mg mL–1) colony were grown at 37 °C and induced by 100 mM isopropyl-b-D-galactopyranoside in the exponential phase (optical density 0.5 at 500 nm). After induction, bacteria were harvested by centrifugation and pellets were resuspended in 5 ml of binding buffer (His-bind buffer kit) (Millipore), proteins were extracted using a Constant cell disrupter system (Constant Systems) with a disruption pressure of 2.35kbar and then purified using His-bind buffer kit (Millipore) following manufacturer's recommendations.

## RNA binding assays

Binding assays were performed using homopolymer RNA conjugated to CNBr-activated Sepharose beads (Sigma **C9142**). Either poly(A), poly(G), poly(C) or poly (U) from Mercks (respectively P9403, P4903, P4404, and P9528) were used following Sigma's recommendations. A total of 0.7–1 mg of RNA were used for 200 mg sepharose beads and 0.45 mg of purified protein (MSIL4 RRM binding domain or mutated version) in 0.5 ml binding buffer (100 mM NaCl, 10 mM Tris-HCl, pH 7.5, 2.5 mM MgCl2, 2.5% Triton X-100) were incubated with 5 mg RNA conjugated Sepharose beads for 1 hr. After incubation, beads were washed five times for 10 min each in 0.5 ml of binding buffer. Beads were then pelleted at 12,000 rpm, resuspended in SDS sample buffer, boiled for 20 min, and pelleted again. Supernatants were transferred to fresh tubes and loaded on a 12% SDS-PAGE minigel for electrophoresis (Bio-Rad) and protein blot analysis.

## RNA immunoprecipitation

RIP was performed as described previously in *Merret et al., 2017* using 400 mg of stem powder as starting material.

## Expression analysis

Total RNA was isolated using the TRI re-agent (Cat. No. TR-118; Euromedex) according to the manufacturer's instructions. Genomic DNA was then digested out using the RQ1 RNase-Free DNase (Cat. No. M6101; Promega). cDNAs were obtained from 400 ng of RQ1-treated RNA using 1 µl of GoScript Reverse Transcriptase (Cat. No. A5003; Promega) in a 20 µl final volume reaction using random primers (Cat. No. C1181; Promega) in the presence of 20 units of RNasin (Recombinant Ribonuclease Inhibitor, Cat. No. C2511; Promega) and dNTP mix at a final concentration of 0.5 mM of each dNTP (Cat. No. U1511; Promega). Semi-quantitative RT–PCR amplifications were performed on 1 µl of cDNA in a 12.5 µl reaction volume to start with. The amplification of EF1αtranscripts was used to equilibrate. qRT–PCR was performed on a Light Cycler 480 II machine (Roche Diagnostics) by using the Takyon No ROX SYBR MasterMix blue dTTP kit (Cat. No. UF-NSMT-B0701; Eurogentec).

Each amplification reaction was set up in a 10 µl reaction containing each primer at 300 nM and 1 µl of RT template in the case of total RNA with a thermal profile of 95 °C for 10 min and 40 amplification cycles of 95 °C, 15 s; 60 °C, 60 s. Relative transcript accumulation was calculated using the ΔΔCt methodology (*Livak and Schmittgen, 2001*), using ACTIN2 as internal control. Average ΔΔCt represents three experimental replicates with standard errors. PolyA +RNAs were purified from RNA obtained by the TRI Reagent extraction, using the PolyATtract mRNA Isolation System III (Cat. No. Z5310; Promega) according to the manufacturer's instructions. cDNAs were then synthetized according to the above protocol using 50 ng of polyA +RNA as starting material. Semiquantitative and quantitative RT–PCRs were performed as described above with a 1/25 dilution of cDNAs from polyA +RNA as template for qRT–PCR. The primers used for RT and qRT-PCR are listed in *Supplementary file 7*.

## RNA sequencing

Total RNA were extracted from the basal stem section of mature Col-0 and *msil2/4* mutant plants using the RNeasy Plant Mini Kit (Qiagen, Cat.#74904) and treated onto the column with the RNase-Free DNase Set (Qiagen, Cat.#79254). Two replicates were performed per plant lines. PolyA purification, library preparation using the Illumina TruSeq stranded mRNA kit and library quality controls were performed by the BioEnvironment Illumina Platform. The mRNA sequencing of Col-0 and msil2/4 mRNAs was done with single-reads, 1x75 pb and 30–35x10$^6$ reads per library.

## RNASeq analysis

For each library, 30–35x10$^6$ reads were obtained with 85–90% of the bases displaying a Q-score ≥30, with a mean Q-score of 38 as assessed with Fastqc (http://www.bioinformatics.babraham.ac.uk/projects/fastqc/). Reads were mapped against the TAIR 10 genome using gtf annotation file and default parameters of TopHat2 v2.0.7 (*Langmead and Salzberg, 2012*). Assembly and transcript quantification were performed with Cufflinks v2.2.1 (*Kim et al., 2013*). Finally, the low expressed transcripts - less than 1 RPKM (reads per kilobase per million mapped reads) in one of the libraries were filtered out. The differential analysis was conducted by Cuffdiff software, belonging to the Trapnell suite. Foldchange (FC) was determined as the ratio between the normalized read counts between mutant and wild-type.

## Fluorescence detection and tissue colorations

GFP detections were performed on in vitro grown one week seedling roots fixed in fixation buffer (50 mM Pipes, 5 mM EGTA, 5 mM MgSO4, pH7 added with 2% para-formaldehyde and 0.2% Triton). After 5 min vacuum infiltration and 10 min agitation at room temperature, roots were washed twice with fixation buffer and stored at 4 °C until confocal observation. Confocal imaging was done using an Zeiss Axio Observer Z1 LSM 700. Roots were observed in water and for GFP detection samples were excited at 488 nm and the signal captured at 495–555 nm. For RFP detection, excitation was performed at 555 nm and the signal captures at 600–700 nm. Images were analysed with ImageJ. At least 15 samples per line were analyzed.

Stem coloration were performed from 5 cm length basal stem pieces conserved in ethanol 80%. Stems were rehydrated first in ethanol 50% and later in water and then embedded in agarose 5% before being transversally cut using a vibratome (Leica vt 1000 s) to obtain 120 µm thickness sections stored in water. The presence of lignin was determined by staining with phloroglucinol for 30 s, giving a red coloration in the presence of lignin cinnamaldehyde groups. Sections were observed using an inverted microscope (Leitz DMRIBE, Leica Micro-systems, Wetzlar, Germany) and images were registered using a CCD camera (Color Collview, Photonic Science, Milham, UK).

For cellulose detection, the brightener Calcofluor White ST [4,4'-bis(anilino-6-bis(2-hydroxyethyl)-amino-s-triazine-ylamino)–2,2'-stilbene disulfonic acid; Cyanamide Co., Bound Brook, N.J] was used at a final concentration of 0.1%. The samples were incubated with calcofluor for 3 min, washed three times with water before being observed. For detection, excitation was performed at 405 nm and the signal captures at 434–496 nm. For auto-fluorescence analysis stem sections were excited at 405 nm and the signal captures at 410–470 nm for blue. Pictures have been analyzed with LAS X (Leica Application Suite X).

## Biochemical analyses of SCWs

*Arabidopsis* stems (5 cm sections at the bottom part of the stems) were harvested from Col0 and *msil2/4* double mutant as four independent biological replicates. Soluble extractives were eliminated as described by *Ployet et al., 2018*. Briefly, stems were freeze dried for 48 hr, ground using a Mixer Mill MM 400 (Retsch), and extracted by hot solvents (successively water, ethanol, ethanol/toluene (1:1 v/v) and acetone) to obtain extractive-free stem residues (ESR). The determination of acetyl bromide (AcBr) lignin was performed on 10 mg of ESR as described by *Ployet et al., 2019*. All analyses were done in triplicate. Fourier transform infrared spectroscopy (FT-IR) analysis was performed on 100–200 mg of ESR. Spectra were recorded from ten technical replicates, in the range 400–4000 cm$^{-1}$ with a 4 cm$^{-1}$ resolution and 32 scans per spectrum using an attenuated total reflection (ATR) Nicolet IS50 FT-IR spectrometer (Thermo Fisher, Illkirch-Graffenstaden, France) equipped with a deuterated-triglycine sulfate (DTGS) detector. Spectra analyses were performed as described in *Dai et al., 2020*. Saccharification yield was estimated with or without alkali pretreatment, as described by *Acker et al., 2016*. Reducing sugar concentration was assessed with dinitrosalicylic acid (DNS) reagent using 10 µl of the supernatant after 9 hr of reaction. Enzyme activity was assessed at 0.25 filter paper units (FPU) ml$^{-1}$ in our conditions. The Boxplots were generated using ggplot2 R package (R v2022.07.1).

## Mass spectrometry (MS)-based proteomic analyses

Proteins from total inflorescence stem extracts of three biological replicates of Col-0 and *msil2/4* mutant plants were solubilized in Laemmli buffer and heated for 10 min at 95 °C. They were then stacked in the top of a 4–12% NuPAGE gel (Invitrogen), stained with Coomassie blue R-250 (Bio-Rad) before in-gel digestion using modified trypsin (Promega, sequencing grade) as previously described (*Casabona et al., 2013*). The resulting peptides were analyzed by online nanoliquid chromatography coupled to MS/MS (Ultimate 3000 RSLCnano and Orbitrap Exploris 480, Thermo Fisher Scientific) using a 180 min gradient. For this purpose, the peptides were sampled on a precolumn (300 µm x 5 mm PepMap C18, Thermo Scientific) and separated in a 75 µm x 250 mm C18 column (Aurora Generation 2, 1.6 µm, IonOpticks). The MS and MS/MS data were acquired using Xcalibur 4.4 (Thermo Fisher Scientific).

Peptides and proteins were identified by Mascot (version 2.8.0, Matrix Science) through concomitant searches against the Viridiplantae database (from Uniprot, *Arabidopsis thaliana* (thale-cress) taxonomy, 136447 sequences) and a homemade database containing the sequences of classical contaminant proteins found in proteomic analyses (human keratins, trypsin..., 126 sequences). Trypsin/P was chosen as the enzyme and two missed cleavages were allowed. Precursor and fragment mass error tolerances were set at respectively at 10 and 20 ppm. Peptide modifications allowed during the search were: Carbamidomethyl (C, fixed), Acetyl (Protein N-term, variable) and Oxidation (M, variable). The Proline software (*Bouyssié et al., 2020*; version 2.2.0) was used for the compilation, grouping, and filtering of the results (conservation of rank 1 peptides, peptide length ≥6 amino acids, false discovery rate of peptide-spectrum-match identifications <1% (*Couté et al., 2020*), and minimum of one specific peptide per identified protein group). MS data have been deposited to the ProteomeXchange Consortium via the PRIDE partner repository (*Perez-Riverol et al., 2022*). Proline was then used to perform a MS1 label-free quantification of the identified protein groups based on razor and specific peptides.

Statistical analysis was performed using the ProStaR software (*Wieczorek et al., 2017*) based on the quantitative data obtained with the four biological replicates analyzed per condition. Proteins identified in the contaminant database, proteins identified by MS/MS in less than two replicates of one condition, and proteins quantified in less than three replicates of one condition were discarded. After log2 transformation, abundance values were normalized using the variance stabilizing normalization (vsn) method, before missing value imputation (SLSA algorithm for partially observed values in the condition and DetQuantile algorithm for totally absent values in the condition). Statistical testing was conducted with limma, whereby differentially expressed proteins were selected using a log2(Fold Change) cut-off of 1 and a p-value cut-off of 0.00513, allowing to reach a false discovery rate close to 1% according to the Benjamini-Hochberg estimator. Proteins found differentially abundant but identified by MS/MS in less than two replicates, and detected in less than three replicates, in the condition in which they were found to be more abundant were invalidated (*P*-value = 1).

## MALDI-TOF MS analyses of the patterns of xylanase-released SCW oligosaccharide

Ten mg of dried alcohol insoluble cell wall residues were incubated with 10 U.ml$^{-1}$ of endo-1,4-D-xylanase (E-XYLNP) from MEGAZYME previously desalted with centrifulgal filters (Amicon Ultra-0.5ml) in 1 ml during 12 hr at 40 °C. After boiling during 5 min to inactivate enzyme activity, samples where filtred with 50KD centrifulgal filters (Amicon Ultra-0.5ml). As positive control, 2 mg of wheat AX (MEGAZYMZE) were treated in a similar manner.

The samples were then analyzed by matrix-assisted laser desorption ionization-time of flight (MALDI-TOF) MS using DMA/DHB (N,N-dimethylaniline/2,5-dihydroxybenzoic acid) matrix (*Ropartz et al., 2011*). The samples (1 µL) were deposited and then covered by the matrix (1 µL) on a polished steel MALDI target plate. MALDI measurements were then performed on a rapifleX MALDI-TOF spectrometer (Bruker Daltonics, Bremen, Germany) equipped with a Smartbeam 3D laser (355 nm, 10,000 Hz) and controlled using the Flex Control 4.0 software package. The mass spectrometer was operated with positive polarity in reflectron mode. Spectra were acquired in the range of 560–3200 *m/z*. The detected species correspond to ions in the form of sodium adducts.

## Thioacidolysis

This method is adapted from *Lapierre et al., 1995*. Briefly, 10 mg of alcohol insoluble cell wall residues were incubated in 3 ml of dioxane with ethanethiol (10%), BF3 etherate (2.5%) containing 0.1% of heinecosane C21 diluted in CH2Cl2 at 100 °C during 4 hr. Three ml of NaHCO3 (0.2 M) were added after cooling and mixed prior the addition of 0.1 mL of HCl (6 M). The tubes were wortexed after addition of 3 mL of dichloromethane and the whole lower organic phase was collected in a new tube before concentration under nitrogen atmosphere to approximately 0.5 ml. Then, 10 µL were trimethylsilylated (TMS) with 100 µL of N,O-bis(trimethylsilyl) trifluoroacetamide and 10 µL of ACS-grade pyridine. The trimethylsilylated samples were injected (1 µL) onto an Agilent 5973 Gas Chromatography–Mass Spectrometry system. Specific ion chromatograms reconstructed at *m/z* 239, 269 and 299 were used to quantify H, G and S lignin monomers respectively and compared to the internal standard at m/z 57, 71, 85.

## Cell wall sugar analysis

Neutral monosaccharide sugar content was determined by gas chromatography after acid hydrolysis and conversion of monomers into alditol acetates as described in *Hoebler et al., 1989*, and *Blakeney et al., 1983*. Gas Chromatography was performed on a DB 225 capillary column (J&W Scientific, Folsorn, CA, USA; temperature 205 °C, carrier gas H$_2$). Calibration was made with standard sugar solution and inositol as internal standard.

For acid sugar analysis, the uronic acid content was automatically quantified after hydrolysis of polysaccharides in concentrated sulfuric acid (18 M) containing sodium tetraborate (12.5 mM) or not, followed by m-hydroxydiphenyl colorimetric determination (*Blumenkrantz and Asboe-Hansen, 1973*; *Thibault, 1979*). Galacturonic and Glucuronic acid were used as standard for quantification.

## Immunohistochemistry

Samples of *Arabidopsis* tissues were dehydrated in a successive ethanol series (20%, 40%, 60%, 80%, 95%, and 100%) and embedded in LR White resin (Electron Microscopy Sciences). Thin sections (1 µm) were placed on Teflon-coated slides (Electron Microscopy Sciences), blocked in phosphate-buffered saline, 2% Tween, and 1% bovine serum albumin for 2 h (PBST-BSA), and labeled overnight (12 hr) at 4 °C with primary antibody diluted in PBST-BSA. Sections were washed with PBST and incubated at room temperature for 2 hr with a secondary antibody diluted in PBST-BSA. Slides were then washed with deionized water and dried under a stream of dry air. Primary antibodies and dilutions were as follows:, LM10 and LM11 (1:10, v/v; Plant Probes for both). The secondary antibody was a goat anti-rat IgG coupled to the fluorescent dye Alexa Fluor 555 (Molecular Probes) and was used at a 1:100 (v/v) dilution. Observations were carried out using a LeicaDM 6000microscope with a TCS SP8 scanning confocal system using a 40 x apochromatic oil immersion objective to visualized Alexa555 fluorescence (excitation: 561 nm; emission: 565–645 nm). Acquisition settings, including laser power,photomultiplier gain, field of view (X,Y dimension), Z-step, and pixel size of the image, were strictly identical to ensure reliable comparisons between plant material (i.e. the wild type versus mutants).

## Acknowledgements

We thank Aurélie Le Ru for the histological staining and cell wall thickness measurements, Hua Cassan-Wang for helping to the preparation of stem sections for staining, and Michele Laudié for preparation of the mRNA libraries. This work was supported by the Centre National de la Recherche Scientific (CNRS), and grants ANR-08-BLAN-0206 and 12-BSV6-0010 from Agence National de la Recherche (ANR) to TL. This study was supported by the "Ecole Universitaire de Recherche (EUR) TULIP-GS (ANR-18-EURE-0019), and the EPIPLANT Groupement de Recherche (CNRS, France). The proteomic experiments performed by PH and DG from the Interdisciplinary Thematic Institute IMCBio, as part of the ITI 2021–2028 program of the University of Strasbourg, CNRS and Inserm, were supported by IdEx Unistra (ANR-10-IDEX-0002), by SFRI-STRAT'US project (ANR 20-SFRI-0012), and EUR IMCBio (IMCBio ANR-17-EURE-0023). The proteomic experiments performed by LB and YC were partially supported by Agence Nationale de la Recherche under projects ProFI (Proteomics French Infrastructure, ANR-10-INBS-08) and GRAL, a program from the Chemistry Biology Health (CBH) Graduate School of University Grenoble Alpes (ANR-17-EURE-0003). The funders had no role in study design, data collection and analysis, decision to publish or preparation of the paper.

## Additional information

### Funding

| Funder | Grant reference number | Author |
| --- | --- | --- |
| Agence Nationale de la Recherche | ANR-08-BLAN-0206 | Thierry Lagrange |
| Agence Nationale de la Recherche | ANR-18-EURE-0019 | Thierry Lagrange |
| Agence Nationale de la Recherche | Proteomics French Infrastructure | Yohann Couté |
| Agence Nationale de la Recherche | ANR-17-EURE-0003 | Yohann Couté |
| Agence Nationale de la Recherche | ANR-10-INBS-08 | Yohann Couté |
| Agence Nationale de la Recherche | ANR-10-IDEX-0002 | Dominique Gagliardi |
| Agence Nationale de la Recherche | 12-BSV6-0010 | Thierry Lagrange |
| Agence Nationale de la Recherche | ANR 20-SFRI-0012 | Dominique Pontier |
| Agence Nationale de la Recherche | IMCBio ANR-17-EURE-0023 | Dominique Pontier |

The funders had no role in study design, data collection and interpretation, or the decision to submit the work for publication.

### Author contributions

Alicia Kairouani, Claire Picart, Remy Merret, Jacinthe Azevedo, Investigation; Dominique Pontier, Formal analysis, Investigation, Methodology, Writing - original draft; Fabien Mounet, Data curation, Formal analysis, Investigation, Visualization, Writing - original draft; Yves Martinez, Mathieu Fanuel, Formal analysis, Investigation, Visualization, Writing - original draft; Lucie Le-Bot, Investigation, Visualization; Philippe Hammann, Data curation, Formal analysis, Investigation; Lucid Belmudes, Data curation, Formal analysis, Investigation, Visualization; Marie-Christine Carpentier, Formal analysis, Investigation; Dominique Gagliardi, Formal analysis, Investigation, Methodology, Writing - review and editing; Yohann Couté, Conceptualization, Data curation, Formal analysis, Methodology, Writing - original draft; Richard Sibout, Conceptualization, Formal analysis, Investigation, Methodology, Writing - original draft; Natacha Bies-Etheve, Conceptualization, Formal analysis, Investigation, Writing

- original draft, Writing - review and editing; Thierry Lagrange, Conceptualization, Data curation, Formal analysis, Investigation, Writing - original draft

## Author ORCIDs
Remy Merret ⓘ http://orcid.org/0000-0002-3790-1115
Yohann Couté ⓘ http://orcid.org/0000-0003-3896-6196
Thierry Lagrange ⓘ http://orcid.org/0000-0003-3090-0918

Reviewer #1 (Public Review): https://doi.org/10.7554/eLife.88207.3.sa1
Reviewer #2 (Public Review): https://doi.org/10.7554/eLife.88207.3.sa2
Reviewer #3 (Public Review): https://doi.org/10.7554/eLife.88207.3.sa3
Author Response https://doi.org/10.7554/eLife.88207.3.sa4

---

## Additional files

### Supplementary files
• Supplementary file 1. Affinity Purification - Mass spectrometry (AP-MS) - based proteomic analyses.

• Supplementary file 2. Wave numbers from FT-IR absorption spectra and their related compounds reported in literature.

• Supplementary file 3. Genes down-regulated in msil2/4.

• Supplementary file 4. MS-based label-free quantitative proteomic analysis of Col-0 (WT) and msil2/4 (mutant).

• Supplementary file 5. Quantification of the LM10 and LM11 fluorescence signals from the xylem cells from Col-0 and msil2/4 mutant inflorescence stems.

• Supplementary file 6. Normalization of Col-0 and msil2/4 xylan MALDI-TOF MS peak intensities using an internal standard (pentaacetyl-chitopentaose).

• Supplementary file 7. Primers used in this work.

• MDAR checklist

### Data availability
Sequencing data have been deposited in GEO under accession code GSE223732.The mass spectrometry proteomics data have been deposited to the ProteomeXchange Consortium under the dataset identifiers PXD040207, and PXD040020.

The following datasets were generated:

| Author(s) | Year | Dataset title | Dataset URL | Database and Identifier |
|---|---|---|---|---|
| Kairouani A, Pontier D, Picart C, Mounet F, Martinez Y, Le-Bot L, Fanuel M, Hammann P, Belmudes L, Merret R, Azevedo J, Carpentier MC, Gagliardi D, Couté Y, Sibout R, Bies-Etheve N, Lagrange T | 2023 | Effect of depletion of MSIL2 and MSIL4 RNA-binding proteins on gene expression in the Arabidopsis inflorescence stem | https://www.ncbi.nlm.nih.gov/geo/query/acc.cgi?&acc=GSE223732 | NCBI Gene Expression Omnibus, GSE223732 |

*Continued*

| Author(s) | Year | Dataset title | Dataset URL | Database and Identifier |
|---|---|---|---|---|
| Kairouani A, Pontier D, Picart C, Mounet F, Martinez Y, Le-Bot L, Fanuel M, Hammann P, Belmudes L, Merret R, Azevedo J, Carpentier MC, Gagliardi D, Couté Y, Sibout R, Bies-Etheve N, Lagrange T | 2023 | Cell type-specific control of secondary cell wall formation by Musashi-type translational regulators in Arabidopsis | https://dx.doi.org/10.6019/PXD040207 | ProteomeXchange Consortium, 10.6019/PXD040207 |
| Kairouani A, Pontier D, Picart C, Mounet F, Martinez Y, Le-Bot L, Fanuel M, Hammann P, Belmudes L, Merret R, Azevedo J, Carpentier MC, Gagliardi D, Couté Y, Sibout R, Bies-Etheve N, Lagrange T | 2023 | Musashi-type translational regulator controls the buildup of secondary cell wall in Arabidopsis | https://dx.doi.org/10.6019/PXD040020 | ProteomeXchange Consortium, 10.6019/PXD040020 |

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
