## [Editor Report · eLife assessment]

Secondary cell walls support vascular plants and conduct water throughout the plant body, and are **important** resources for lignocellulosic feedstocks. Here the authors present **convincing** genetic and biochemical evidence that secondary cell wall synthesis, known already to be under complex transcriptional control, is also controlled post-transcriptionally by MUSASHI-like RNA-binding proteins. These **important** results point to a new mechanism for control of secondary cell wall synthesis, which will be interesting to cell biologists and biochemists studying and attempting to manipulate plant biomass.

---

## [Referee Report · Reviewer #1 (Public Review)]

Secondary cell walls support vascular plants and conduct water throughout the plant body, but are also important resources for lignocellulosic feedstocks. Secondary cell wall synthesis is under complex transcriptional control, presumably because it must only be initiated after cell growth is complete. Here, the authors found that two Musashi-type RNA-binding proteins, MSIL2 and MSIL4 are redundantly required for secondary cell wall development in Arabidopsis. The plant phenotypes could be complemented by the wild-type version of either protein, but not by a MSIL4 version that carries mutations in the conserved RNA-binding domains, and the authors localized MSIL2 & 4 to stress granules, implicating the RNA-binding function of MSIL4 in the cell wall phenotype. Upon closer inspection, the secondary cell wall phenotypes included changes in vasculature morphology, and minor changes to lignin and hemicellulose (glucuronoxylan). While there were no changes to likely cell wall target genes in the transcriptome of msil2msil4 plants, proteomics experiments found glucuronoxylan biosynthesis components were upregulated in the mutants, and they detected an increase in substituted xylan via several methods. Finally, they documented MSIL4 binding to RNA encoding one of these targets, suggesting that MSIL2 and MSIL4 act to post-transcriptionally regulate glucuronoxylan modification. Altogether, this is a new mechanism by which cell wall composition could be regulated.

Overall, the manuscript is well-written, the data are generally high-quality, and the authors typically use several independent methods to support each claim. However, several important questions remain unanswered by this work in its current state and the model presented in Figure 7 is quite speculative. For example, the link between the striking plant phenotype and GXM misregulation is unclear since GXM overexpression doesn't alter plant phenotypes or lignin content (Yuan et al 2014 Plant Science), so misregulation of GXMs in msil2msil4 mutants clearly is not the whole story. It also remains to be determined why one particular secondary cell wall synthesis enzyme is regulated likely post-transcriptionally, while so much of the pathway is regulated at the transcriptional level. There are likely other targets for MSIL2- and MSIL4-mediated regulation since it seems that MSIL2 and MSIL4 are expressed in tissues that are not synthesizing secondary cell walls.

---

## [Referee Report · Reviewer #2 (Public Review)]

This work explored the biological functions of a small family of RNA-binding proteins that was previously studied in animals, but was uncharacterized in plants. Combinatorial T-DNA insertional mutants disrupting the expression of the four Mushashi-like (MSIL) genes in Arabidopsis revealed that only the msil2 msil4 double mutant visibly alters plant development. The msil2/4 plants produced stems that could not stand upright. Transgene complementation, site-directed mutagenesis of MSIL4 conserved RNA-binding motifs, and in vitro RNA binding assays support the conclusion that the loss of MSIL2 and MISL4 function is responsible for the observed morphological defects. MSIL2/4 interact with proteins associated with mRNA 3'UTR binding and translational regulation.

The authors present compelling biochemical evidence that Mushashi-like2 (MSIL2) and MSIL4 jointly regulate secondary cell wall biosynthesis in the Arabidopsis stem. Quantitative analyses of proteins and transcripts in msil2/4 stems uncovered transcriptional upregulation of several xylan-related enzymes (despite WT-like RNA levels). Consistent with MALDI-TOF data for released xylan oligosaccharides, the authors propose a model in which MSIL2/4 negatively regulate the translation of GXM (glucuronoxylan methyltransferase), a presumed rate-limiting step. The molecular links between overmethylated xylans and the observed stem defects (which include subtle reductions in lignin and increases beta-glucan polymer distribution) warrants further investigation in future studies. Similarly, as the authors point out, it is intriguing that the loss of the broadly expressed MSIL2/4 genes only significantly affects specific cell types in the stem.

---

## [Referee Report · Reviewer #3 (Public Review)]

The manuscript by et al. investigates the function of a small family of plant RNA binding proteins with similarity to the well-studied Musashi protein in animals, and, therefore, called MUSASHI-LIKE1-4 (MSL1-4). Studies on the biological importance of post-transcriptional control of gene expression via RNA-binding proteins in plants are not numerous, and advances in this important field are much needed. The thorough work presented in this manuscript is such an advance.

The central observations of the paper are

- Knockout of any MSL gene alone does not produce a phenotype.

It is of note that basic characterization of knockout mutations is really well done - for example, the authors have taken care to raise specific antibodies to each of the MSL proteins and use them to demonstrate that each of the T-DNA insertion mutants used actually does knock out protein production from the corresponding gene.

- Knockout of MSL2/4 (but no other double mutant) produces a clear leaf phenotype, and a remarkable stem phenotype in which the mutants collapse as they are unable to support upright growth

- The phenotypes of knockout mutants persist in point mutants defective in RNA-binding, indicating that RNA-binding is required for biological activity. Consistent with this, and associate physically with other RNA-binding proteins and translation factors.

- MSL proteins are cytoplasmic

- The msl2/4 mutants present multiple defects in secondary cell wall composition and structure, probably explaining their inability to grow upright. I did not examine the cell wall analyses in detail as I am no specialist in this field.

- Msl2/4 mutants show transcriptomic changes with at large two big categories of differentially expressed genes compared to wild type.

(1) Genes related to cell wall metabolism

(2) Genes associated with defense against herbivores and pathogens

- Two of the mRNAs encoding cell wall factors with significant upregulation in msl2/4 mutants compared to wild type also associate physically with MSL4 as judged by RNA-immunoprecipitation-RT-PCR assays, and this physical association is abrogated in the RNA-binding deficient MSL4 mutant.

Altogether, the study shows clear biological relevance of the MSL family of RNA-binding proteins and provides good arguments that the underlying mechanism is control of mRNAs encoding enzymes involved in secondary cell wall metabolism (although concluding on translational control in the abstract is perhaps saying too much - post-transcriptional control will do given the evidence presented). One observation reported in the study makes it vulnerable to alternative interpretation, however, and I think this should be explicitly treated in the discussion:

The fact that immune responses are switched on in msl2/4 mutants could also mean that MSL2/4 have biological functions unrelated to cell wall metabolism in wild type plants, and that cell wall defects arise solely as an indirect effect of immune activation that is known to involve changes in expression of many cell wall-modifying enzymes and components such as pectin methylesterases, xyloglucan endotransglycosylases, arabinogalactan proteins etc. Indeed, the literature is rich in examples of gene functions that have been misinterpreted on the basis of knockout studies because constitutive defense activation mediated by immune receptors was not taken into account (see for example Lolle et al., 2017, Cell Host & Microbe 21, 518-529).

With the evidence presented here, I am actually close to being convinced that the primary defect of msl2/msl4 mutants is directly related to altered cell wall metabolism, and that defense responses arise as a consequence of that, not the other way round. But I do not think that the reverse scenario can be formally excluded with the evidence at hand, and a discussion listing arguments in favor of the direct effect proposed here would be appropriate. Elements that the authors could consider to include would be the isolation of a cellulose synthase mutant as a constitutive expressor of jasmonic acid responses (cev1) as a clear example that a primary defect in cell wall metabolism can produce defense activation as secondary effect. The interaction of MSL4 with GXM1/3 mRNAs is also helpful to argue for a direct effect, and it would strengthen the argument if more examples of this kind could be included.

---

## [Author Response]

The following is the authors’ response to the original reviews.

Please find below our detailed point-by-point response to the eLife reviewer comments. As suggested by the reviewers, we have (1) replaced most of the Bar charts by Box plots, (2) highlighted the sucellular regions that are analyzed in the measurement experiments, and (3) have rewritten and toned down several subsections of the discussion.

**Reviewer #1 (Recommendations For The Authors):**
I suggest that the authors consider the following points in future versions of this manuscript:1). The link between the striking plant phenotype and GXM misregulation is unclear since GXM overexpression doesn't alter plant phenotypes or lignin content (Yuan et al 2014 Plant Science), so misregulation of GXMs in msil2msil4 mutants clearly is not the whole story. The authors should discuss alternative interpretations of their results and other possible targets of MSIL2/4 that might be contributing to the plant phenotype.

We completely agree with the reviewer that the misregulation of GXMs in msil2/4 is not the whole story and we are currently developing specific strategies in order to characterize in an unbiased manner the full repertoire of MSIL mRNA targets in the stem, hoping we can identify other targets relevant to the formation of SCW. We have also toned-down our discussion concerning the possible impact of glucuronoxylan methylation level on lignin deposition (L546-552).

1. Similarly, it remains unclear why one particular secondary cell wall enzyme is regulated post-transcriptionally, while so much of the pathway is regulated at the transcriptional level. Please discuss.

We do not exclude that other genes encoding for SCW enzymes are impacted and it will be the subject of further investigations. We have extended the discussion concerning these points. We have extended the discussion concerning these points (L486-498).

1. Thirdly, it seems that MSIL2 and MSIL4 are expressed in tissues that are not synthesizing secondary cell walls. The authors should discuss other possible targets of MSIL2/4 from their work.

We have extended the discussion concerning the pleiotropic effects of MSIL mutation in Arabidopsis (L 416-425). The variability of the msil2/4 phenotype is so large that we expect these proteins to regulate various cellular functions through the binding of specific set of mRNA. The mRNA targets specifically involved in these regulations will need to be determined on a case-by-case basis.

1. The discussion is extremely speculative and introduces new abbreviations (LTAc, XTRe) that are only used in their model (Figure 7). I suggest replacing these with dashed lines and/or question marks in the model, since as currently depicted, it looks as if these could be known gene products, which could be very misleading.

We have removed the Ltac and XTRe abbreviations in Figure 7, and the corresponding text in the discussion section.

1. Similarly, the speculation that cellulose content somehow regulates glucuronoxylan levels via xylan-cellulose interactions, leading to degradation of excess glucuronoxylan after synthesis is, to my knowledge, completely unsupported by any evidence except the correlation between cellulose and xylan levels. Please either support this claim with references or remove it from the discussion.

We have removed the claim and have rewritten and toned down the text accordingly to the reviewer 1 comments (L 499-512).

1. Bar charts are rarely the most appropriate method for displaying biological data (Streit & Gehlenborg 2014 Nature Methods). Authors should replace bar charts with one of the following options: (A) plot all individual datapoints and overlay summary statistics, (B) box plots with all individual datapoints show, (C) violin plots (when n is large, i.e. n > 50). R and R studio are free software that can generate such plots. Several excellent tools exist online to generate such plots via a free, graphical user interface, such as boxplotr (Spitzer et al 2014 Nature Methods): http://shiny.chemgrid.org/boxplotr/ and PlotsOfData (Postma & Goedhart 2019 PLoS Biology): https://huygens.science.uva.nl/PlotsOfData/

We have replaced the Bar charts in figure 4E,G and Fig 5E with Box plots and acknowledged the software used in the corresponding Materials and methods section.

**Reviewer #2 (Recommendations For The Authors):**
Minor points:Which cells from Fig. 4b were measured for 4c? Some highlighted annotations to delineate the regions that were measured would help.

We have highlighted in figure 4B the subcellular regions cells analyzed in the measurement experiments.

In line 254, the phrase "not merely affected" in the mutant should be rephrased for clarity

We have replaced “not merely affected” by “not significantly” (L274).

Line 317: "we first performed glycome profiling", the data shows monosaccharide profile, not glycome profiling usually involving antibodies microarrays

We have corrected the text according to the reviewer comment (L339-340).

Reviewer #3 (Recommendations For The Authors):Altogether, the study shows clear biological relevance of the MSL family of RNA-binding proteins, and provides good arguments that the underlying mechanism is control of mRNAs encoding enzymes involved in secondary cell wall metabolism (although concluding on translational control in the abstract is perhaps saying too much - post-transcriptional control will do given the evidence presented). One observation reported in the study makes it vulnerable to alternative interpretation, however, and I think this should be explicitly treated in the discussion:The fact that immune responses are switched on in msl2/4 mutants could also mean that MSL2/4 have biological functions unrelated to cell wall metabolism in wild type plants, and that cell wall defects arise solely as an indirect effect of immune activation that is known to involve changes in expression of many cell wall-modifying enzymes and components such as pectin methylesterases, xyloglucan endotransglycosylases, arabinogalactan proteins etc. Indeed, the literature is rich in examples of gene functions that have been misinterpreted on the basis of knockout studies because constitutive defense activation mediated by immune receptors was not taken into account (see for example Lolle et al., 2017, Cell Host & Microbe 21, 518-529).With the evidence presented here, I am actually close to being convinced that the primary defect of msl2/msl4 mutants is directly related to altered cell wall metabolism, and that defense responses arise as a consequence of that, not the other way round. But I do not think that the reverse scenario can be formally excluded with the evidence at hand, and a discussion listing arguments in favor of the direct effect proposed here would be appropriate. Elements that the authors could consider to include would be the isolation of a cellulose synthase mutant as a constitutive expressor of jasmonic acid responses (cev1) as a clear example that a primary defect in cell wall metabolism can produce defense activation as secondary effect. The interaction of MSL4 with GXM1/3 mRNAs is also helpful to argue for a direct effect, and it would strengthen the argument if more examples of this kind could be included.

In accordance to Rev3 comments, we have extended the discussion, listing the arguments, that we believe, are not in favor of a primary effect of the MSIL2/4 proteins on the activation of plant defense pathways (L468-485).

SUGGESTIONS FOR IMPROVED ANALYSES & MINOR TEXT AND FIGURE CORRECTIONS.(1) Unless there is a very good reason to use homology modelling such as SWISS-MODEL (for example ligand-bound proteins), Alphafold2 is now the tool to use for structure prediction. I would at least verify that Alphafold agrees with SWISS-MODEL on the predicted structures shown in Fig 2a.

We have analyzed the MSIL4 sequence using the Alphafold2 prediction software and the output of this analysis completely agrees with the SWISS-Model prediction. We have added an additional panel showing the Alphafold 2 prediction (see figure 2-figure supplement 1B).

(2) The plant pictures shown in Figure 2d are not publication quality in terms of resolution, mounting, size. They really should be redone before final publication.

We thank the reviewer for this important observation, and have improved the resolution of the figure 2D.

(3) The colocalization in Figure 3d/e would benefit from some statistical analysis of the data: How many foci were examined? How many showed colocalization? Is that fraction statistically significant? It can be done from the images at hand; I do not think that additional data acquisition is necessary.

We have used an ImageJ plugin to perform colocalization analysis on the microscopy images corresponding to the bottom panel of the figure 3D (heat stress). This analysis confirmed that most of the foci are actually colocalizing (see Author response image 1). However our initial image data acquisition do not allow us to perform statistical analysis on it. We have added a sentence indicating that colocalization is supported by an analysis using an ImageJ plugin.

**Author response image 1. sa4fig1:** 

1. Typographical and other writing errors:Line 72 "prior to"Line 77 "in the Arabidopsis model"Line 97 "RBP-mediated..."Line 110 "aspects of development"Line 128 "little is known" (no yet)Line 253 "Col-0"Line 346 "previous"

All the writing errors have been corrected in the revised version.